



# A gridded surface current product for the Gulf of Mexico from consolidated drifter measurements

Jonathan M. Lilly[1] and Paula Pérez-Brunius[2]

[1]Theiss Research, La Jolla, California, USA
[2]Departamento de Oceanografía, Centro de Investigación Científica y de Educación Superior de Ensenada (CICESE), Ensenada, Mexico

**Correspondence:** Jonathan Lilly (j.m.lilly@theissresearch.org)

**Abstract.** A large set of historical surface drifter data from the Gulf of Mexico—3761 trajectories spanning 27 years and more than a dozen data sources—are collected, uniformly processed and quality controlled, and assimilated into a spatially and temporally gridded dataset called GulfFlow. This dataset is available in two versions, with one-quarter degree or one-twelfth degree spatial resolution respectively, both of which have overlapping monthly temporal bins with semimonthly spacing, and extend from the years 1992 through 2019. Together these form a significant resource for studying the circulation and variability in this important region. The uniformly processed historical drifter data interpolated to hourly resolution from all publicly available sources are also distributed in a separate product called GulfDriftersOpen. Forming a mean surface current map by directly bin-averaging the hourly drifter data is found to lead to severe artifacts, a consequence of the extremely inhomogeneous temporal distribution of the drifters. Averaging instead the already monthly-averaged data in GulfFlow avoids these problems, resulting in the highest-resolution map of the mean Gulf of Mexico surface currents yet produced. The consolidated drifter dataset is freely available from https://doi.org/10.5281/zenodo.3985916 (Lilly and Pérez-Brunius, 2020a), while the gridded products are available for noncommercial use at https://doi.org/10.5281/zenodo.3978793 (Lilly and Pérez-Brunius, 2020b), the latter being freely available for noncommercial use only for reasons discussed herein.





# 1 Introduction

In addition to being home to a diverse ecosystem, the Gulf of Mexico is vital to the economic interests the United States, Mexico, and Cuba. In order to effectively and safely make use of the Gulf's resources, while doing so in a way that minimizes the risks to the environment, an accurate understanding of the surface currents is necessary.

The Deepwater Horizon catastrophe underscored the urgent need to understand transport and dispersion in the Gulf. In its wake, several major funding initiatives propelled an enormous increase in the scientific activity in this region, with perhaps half

of the studies of the oceanography in the Gulf of Mexico occurring in the past decade. Recent work has focused on topics as diverse as submesoscale dispersion (Poje et al., 2014), mesoscale eddy activity (Le Hénaff et al., 2014), ecosystem health (Joye et al., 2016), coherent structures (Miron et al., 2017), deep circulation (Pérez-Brunius et al., 2018), and cross-shelf transport (Thyng and Hetland, 2018), to name only a few. Such scientific and social importance points to the need to have easy-to-use, well-documented products for studying the Gulf circulation.

Largely because of its economic importance, the circulation in the Gulf has been the subject of a large number of studies carried out by numerous investigators. Among these one may specifically note remotely-tracked surface drifter experiments, which beginning in the 1990s opened a new window into the Gulf surface currents. Surface drifter measurements are unique in their ability to resolve both small spatial scale, fast timescale motions as well as large-scale, long timescale variability. While satellite altimeter-derived maps of the surface geostrophic currents are an invaluable resource with unprecedented spatial and

temporal continuity, their spatial resolution cannot match those derived from of dense surface drifter deployments. Whereas satellite altimetry employs an $O(100)$ km smoothing scale, surface drifters can resolve fluctations as small as hundreds of meters or even meters depending on the tracking method employed.

These drifter experiments have been previously exploited for their scientific content on an individual basis. However, they retain great latent value as possible components of an aggregate. As the various datasets have complementary spatial and

temporal distributions, there is much to be gained by combining them. Other authors, e.g. Miron et al. (2017), Gough et al. (2019), and Mulet et al. (2020), have compiled merged datasets similar to the one created here, and successfully employed these for their own purposes. Yet there currently exists no publicly distributed merged data product derived from the Gulf of Mexico surface drifters. Indeed, notwithstanding the global study of Laurindo et al. (2017) using exclusively Global Drifter Program drifters, no mean circulation maps for the Gulf of Mexico from a drifter-derived dataset have appeared in the literature

since perhaps DiMarco et al. (2005) and Nowlin et al. (2001), at a time when the data coverage was a fraction of what it is today.

Accessing the information content of the historical drifter observations in the Gulf of Mexico is challenging for a number of reasons. To begin with, many of these experiments were carried out before data sharing and archiving practices had evolved into their current efficient and rigorous form. Consequently, the investigator wishing to make use of datasets that appear in

the literature has to first track down this data by navigating government or institutional archiving sites with varying degrees of user-friendliness, as well as obscure individual project sites, or in some cases by personally contacting the investigators. The





data products one then collects are generally presented in a range of custom ascii formats that one must write custom code to read.

After this, one is faced with the task of combining a heterogeneous group of datasets having different sample rates, states of
processing, and physical drifter designs, and having been subjected to a variety of upstream quality control, interpolation, and filtering procedures. Some datasets are distributed with extensive metadata including error estimates from the interpolation, while others include only interpolated latitude and longitude values with no indication of how these relate to the original position fixes. Some are on temporal regular grids with no gaps, some are on regular grids with occasional gaps, and one is presented in its raw, ungridded form. Thus the challenge to the analyst is to combine this information in a sensible way. One
wishes to keep the metadata in the situations when it is available, and when it is not available one would like to use subjective and objective means to identify suspicious or problematic intervals.

The societal importance motivating studies of the Gulf of Mexico has a flip side that one must also contend with. Owing to economic interests in the region, some important datasets are proprietary and are not publicly available. By far the largest is that belonging to Horizon Marine, part of the Woods Hole Group, and described by e.g. Anderson and Sharma (2008) and
Sharma et al. (2010). This dataset was recently used by Mulet et al. (2020) together with several other drifter datasets to create an experimental product of daily velocity maps from merged altimetry and drifter measurements. Unfortunately, this product is only available for a brief eight-month time period, from 01/09/2015 to 30/04/2016. As a part of this project we have obtained access to a subset of the Horizon Marine dataset, as well as to a second proprietary dataset, the Southern Gulf of Mexico (SGOM) dataset owned by the state-owned Mexican oil company Petróleos Mexicano (Pemex) and discussed subsequently.

In order that other researchers may benefit from the valuable set of historical drifter measurements in the Gulf, a spatially and temporally gridded velocity product is created. This product includes data from the proprietary datasets that are available to us, working around the nonnegotiable constraint that the trajectories themselves are not distributable. This data product, called GulfFlow, is intended to facilitate studies of the mean circulation and its interannual and seasonal variability. It contains all velocities from all data sources bin-averaged into either one-quarter or one-twelfth degree spatial bins, and overlapping one
month long temporal bins spaced every half month from August 1992 until July 2019. The number of velocity data points from each of the source datasets contributing to each bin in the 3D spatiotemporal grid is recorded. GulfFlow is freely distributed for noncommercial use, as described in Sect. 7.

The gridded datasets have the advantage that high-quality mean flow estimates are readily obtained through averaging over their time dimension. By contrast, averaging drifter data from all times into spatial bins leads to a distorted mean flow map.
This is a result of the highly inhomogeneous temporal sampling biasing the maps differently at different spatial locations. The improvement in performance due to the two-step temporal averaging can be quantified by applying the same averaging methods to altimetrically inferred velocities and to velocities from numerical models of the region, sampled along the locations of the observed drifter trajectories.

In addition to these two gridded datasets, a merged hourly dataset, called GulfDrifters, is created. This contains quality
controlled and uniformly processed versions of all available surface drifter datasets from the Gulf of Mexico. Drifter locations, velocities derived from these, a bad or missing data flag created during this processing, and metadata from the original sources

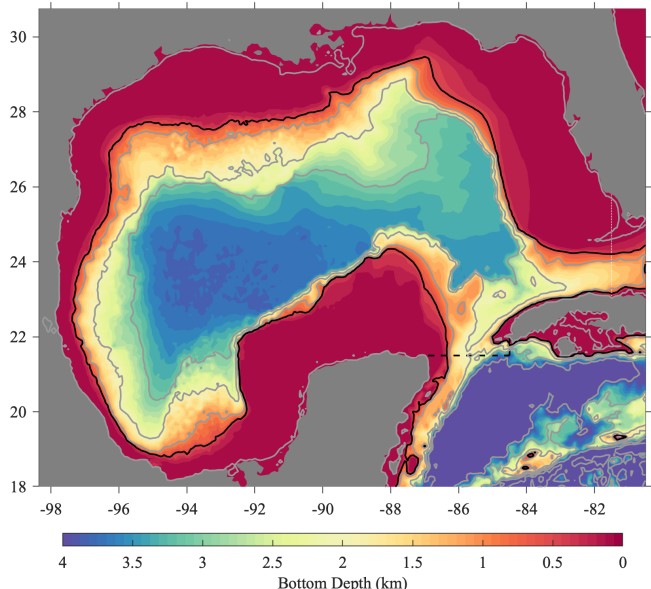

**Figure 1.** Bottom depth in the Gulf of Mexico, in kilometers. The region within which drifter trajectories are extracted is bounded by 80.5° W to the east, the right-hand edge of this plot, and in the Yucatán channel by the dotted line extending east from the Yucatán Peninsula along 21.5° N and turning north to Cuba at 84.5° W. The heavy black contour is the 500 m isobath, while the gray contours mark the 5 m isobath as well as those at 1 km, 2 km, etc. Subsequent plots will extend eastward only to 81.5° W, the dashed white line reaching from Florida to Cuba.

when available, are all incorporated as a part of this product, including in some cases error estimates and drogue presence flags. It is created in two versions, GulfDriftersAll that is the basis for GulfFlow, and GulfDriftersOpen that has three proprietary dataset removed and that is distributed without restriction, see Sect. 7.

For reference, the bathymetry of the Gulf of Mexico is shown in Fig. 1, together with a delineation of the study region. The Gulf is characterized by a broad shelf with depths of 500 m or less, nearly encircling a deep basin with depths as great as 3.5–4 km. The shelf system consists of the Yucatán Shelf (also known as the Campeche Bank) to the south, the Texas-Louisiana Shelf to the north, and the West Florida Shelf to the east. On the eastern side, relatively shallow sills at the Yucatán Channel and the Florida Straits provide narrow openings for the entrance and exit of the poleward flowing Loop Current. These two

straits provide natural cutoffs for the study region. Here we define the Gulf of Mexico to be bounded to the east by the line 80.5° W, and to the south by the line extending eastwards from the Yucatán at 21.5 ° N, then turning north toward Cuba at 84.5° W.

The structure of the paper is as follows. The most fundamental result from this study, an improved surface current map, is presented in Sect. 2 and compared with the best currently available products. The various data sources are described in detail

in Sect. 3, with the processing steps for creating the merged dataset presented in Sect. 4. Special attention is given to possible error and bias sources associated with this heterogeneous dataset. The construction of the gridded dataset is accomplished in



Sect. 5, and errors associated with the creation of the mean flow map are addressed. Conclusions are given in Sect. 6, and data availability is discussed in Sect. 7. Finally, Appendix A gives some details of the numerical processing with reference to a freely available software package for data analysis maintained by the first author.

## 100   2   A improved surface current map

The best currently available estimated time-mean surface current maps for the Gulf of Mexico, created from two very different sources, are shown in Fig. 2 together with two maps created in this paper. All of these velocity maps are on the same quarter-degree grid.

The mean surface currents over the time period January 1, 1993 until May 13, 2019 from a satellite altimeter-derived
velocity product are shown in Fig. 2a. This product is distributed by the Copernicus Marine Environment Monitoring Service (CMEMS), and is essentially the same product that was previously distributed by the Archiving, Validation and Interpretation of Satellite Oceanographic Data (AVISO) service. It involves a substantial amount of spatial smoothing—the result of an optimal interpolation—with zonal smoothing scales of around 170 km at the latitude of the Gulf of Mexico (see Fig. 4a of Pujol et al., 2016).

The Gulf of Mexico portion of the global time-mean surface currents from the climatology produced by Laurindo et al. (2017) is shown in Fig. 2b. This product, created by N OAA's Atlantic Oceanographic and Meteorological Laboratory (AOML) using data from Global Drifter Program (GDP) drifters, will be referred to as the Near-Surface Velocity Climatology or NSVC. Drifter velocities, corrected for slip bias in the case of drogue loss, are subjected to a spatiotemporal fit for all GDP data points within a radius equivalent to one degree of longitude, or about 100 km at these latitudes. Thus, like the altimetry product, this
map involves a spatial smoothing.

In both of these maps, the Loop Current is plainly visible and appears similar in size, shape, and magnitude. A cyclonic gyre, known as the Campeche Gyre (Padilla-Pilotze, 1990; Pérez-Brunius et al., 2013), is seen in the southwestern Gulf of Mexico in Fig. 2b, but is only very faintly present in Fig. 2a, presumably due to the larger smoothing scales in the altimeter product. A southward current near the 500 m isobath off the coast of Florida is seen in both products, although it is stronger
in the drifter-derived product. A northward current occurs near the western edge of the Gulf in the drifter product that is only barely apparent in the altimeter product. These are the major features of note that can be seen in the currently available velocity products.

Directly bin-averaging all good velocity points from the consolidated drifter dataset created here yields the map shown in Fig. 2c. While we see much new detail in the western coastal current and the Campeche Gyre, this map is obviously
unsatisfactory in the region of the Loop Current. It is shown in Sect. 5 that its distorted appearance arises as a consequence of the extremely inhomogeneous distribution of the data over time. This naïve method of averaging overemphasizes the state of the Gulf currents during densely sampled time periods of several large experiments, severely biasing the results toward different time periods in different regions.

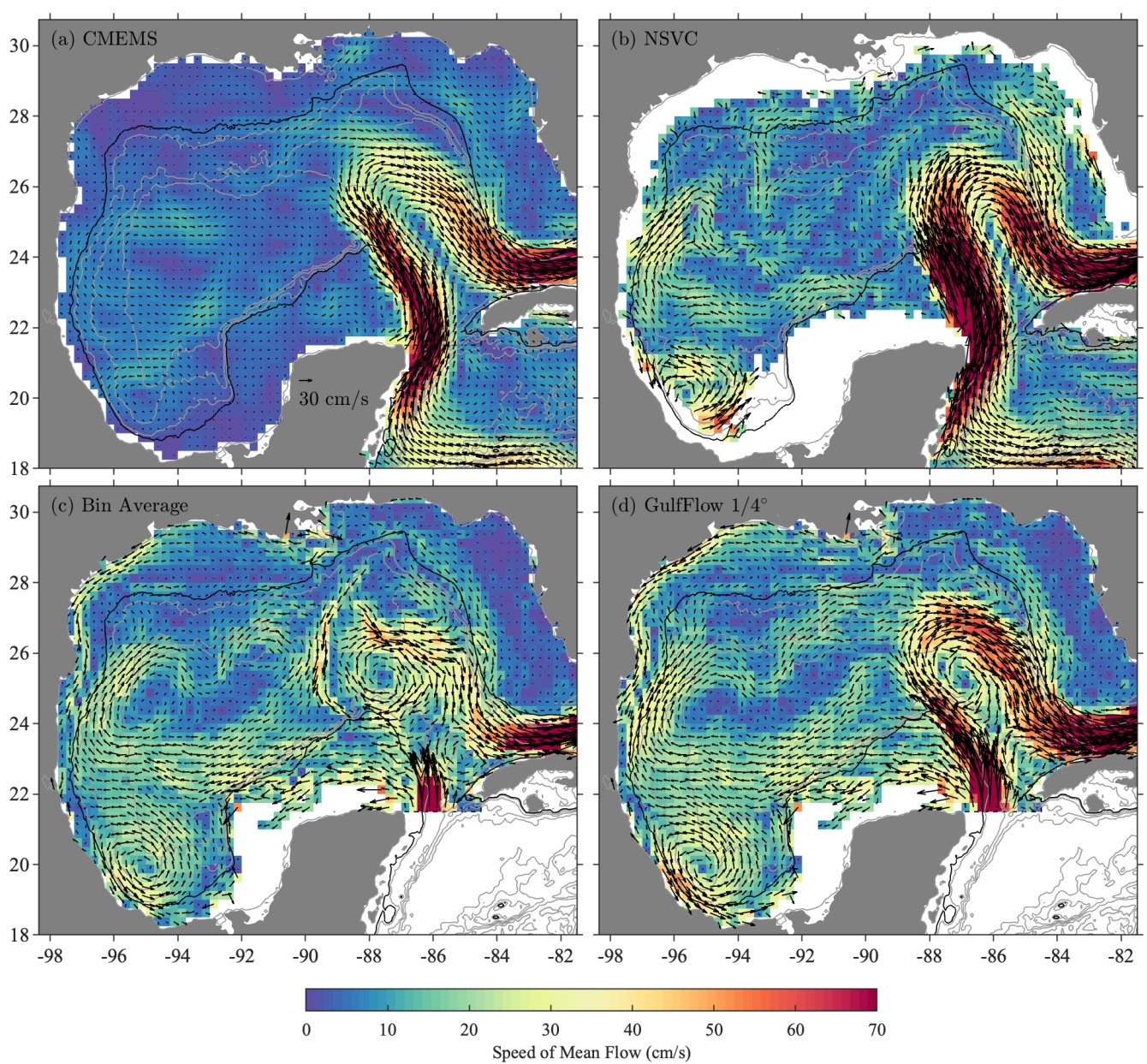

**Figure 2.** The mean surface circulation in the Gulf of Mexico in quarter-degree bins from (a) CMEMS satellite altimetry, (b) the drifter-based climatology of Laurindo et al. (2017), (c) a direct bin-averaging of all data in the GulfDriftersAll dataset, and (d) the time mean of monthly data from the GulfFlow-$1/4°$ product, equivalent to a two-step temporal averaging of the GulfDriftersAll dataset. The colored shading gives the speed of the time-mean flow, also proportional to the length of the arrows. The scale for the arrows is shown in panel (a). For presentational clarity, arrows are shown on a decimated three-quarter degree grid. Bathymetric contours in this and following plots are as in Fig. 1.





This problem is addressed by a two-step averaging procedure. For this we use the GulfFlow-$1/4°$ product created here, which has all drifter data averaged in quarter-degree bins and overlapping monthly bins spaced every half month for twenty-seven years. Averaging over the temporal bins leads to the map shown in Fig. 2d. The sampling artifacts affecting the direct bin average in Fig. 2c have been satisfactorily resolved. The apparently superior performance of this averaging method, while impossible to assess directly from observations, can be estimated by applying the same sampling and averaging schemes to both the CMEMS altimeter fields as well as to the output of several high-resolution numerical models. This is done in Sect. 5, in which we find the estimated reductions in error to be in the range of 32–44%.

Note that the average over overlapping time bins is virtually identical to averaging over only all whole month bins. The semi-monthly temporal spacing is chosen such that seasonal variability can be better resolved, so we average over all temporal bins for convenience.

Unlike the first row of Fig. 2, the second row involves no spatial averaging apart from the quarter-degree bin-averaging. Consequently, features are seen at much higher resolution. A southward-flowing coastal current is revealed, extending from Louisiana to about 24° N, that is entirely absent in Figs. 2a,b. The northward flowing shelf-break current near 24° N from Fig. 2b takes on a more eddy-like or gyre-like shape in Fig. 2d. At its northern edge, a bifurcation is seen where part of the mean current turns to the north while part of it turns to the east. A large-scale, bean-shaped anticyclonic circulation, with a pronounced velocity minimum along its center, is seen extending throughout the deep Gulf from west of the Loop Current to the western coast. In short, a number of apparently physically meaningful features are seen that cannot be discerned in currently available products.

This velocity map can be improved still further. A higher resolution version of the GulfFlow product, GulfFlow-$1/12°$, is created with $1/12°$ spatial binning instead of $1/4°$. All monthly data are then smoothed using a local parabolic weighting function, $1 - r^2/R^2$, that decays to zero at a radius of $R =$50 km and that is zero outside of that radius. The resulting mean flow estimate is shown in Fig. 3a.

Comparing the smoothed $1/12°$ map in Fig. 3a with the $1/4°$ binned map in Fig. 2d, we see that the former has more detail in regions of finescale structure such as the counter-flowing currents of the western boundary current, the Mississippi outflow plume, and the interior of the Loop Current. In particular, the Mississippi outflow region in Fig. 3a clarifies the apparently nonsensical jumble of vectors seen in that region in Fig. 2d. What is happening in the $1/4°$ map is that the grid is not fine enough to resolve the plume structure, leading to vectors in adjacent bins that seem unrelated to one another. When the grid is fine enough to resolve the structure, the same data leads to the meaningful outflow pattern seen in Fig. 3a. High-resolution modeling studies such as that of Barkan et al. (2017) also show strong, narrow outflow plumes in the region, see their Fig. 8.

The streamlines corresponding to the $1/12°$ mean flow map, in Fig. 3b, accentuate the closed circulations in the Campeche Gyre and in the central and western deep Gulf. The closed circulation over the Loop Current likely in part reflects a bias of the dataset towards a state in which an eddy is present or separating due to intentional launching of drifters from one of the data sources within eddies, as discussed later. This figure also reveals the robust east/west connectivity present over the deep part of the Gulf, with streamlines reaching over some ten degrees of longitude, as well as the strong north-south connectivity along the western boundary.



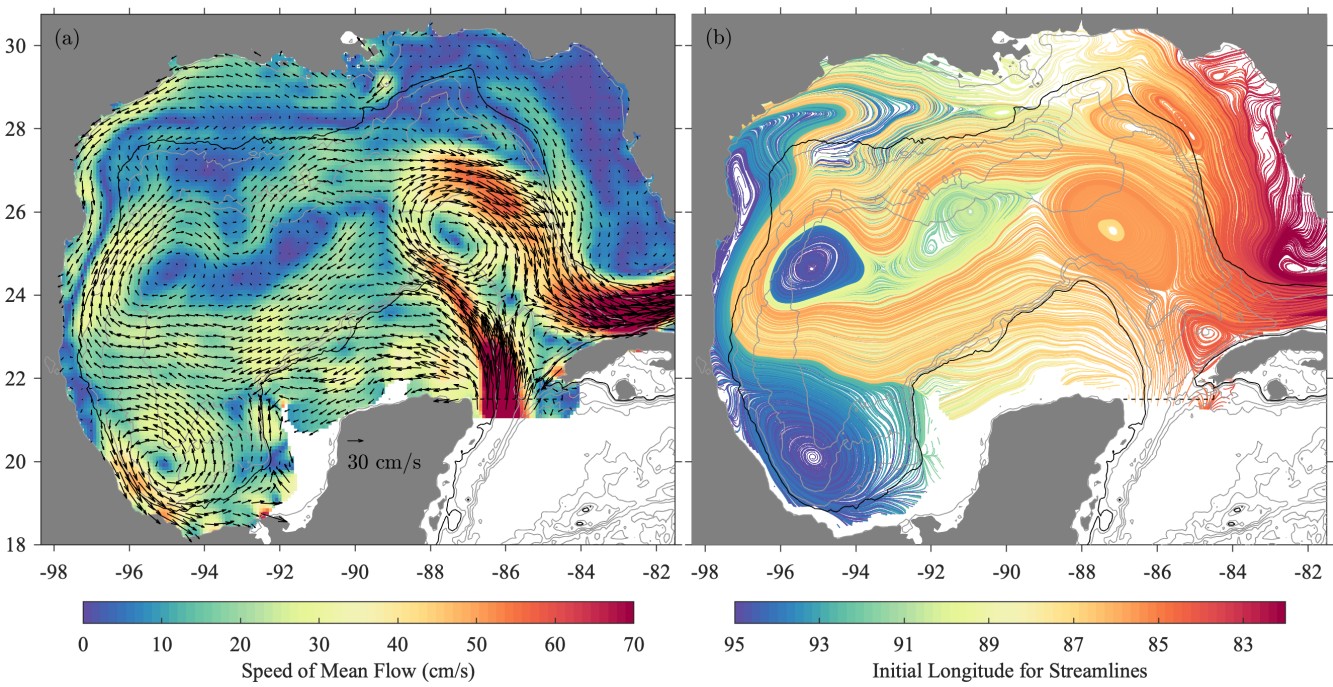

**Figure 3.** Panel (a) is the surface circulation in the Gulf of Mexico as in Fig. 2, but for the GulfFlow-1/12° product smoothed within 50 km radius circles as described in the text. Panel (b) shows streamlines of the mean flow in (a), colored according to their initial longitude.

The maps in Fig. 3 represent the highest resolution estimate of the mean Gulf of Mexico surface currents created to date.
Further examination of the patterns seen here, as well as of temporal variability captured by the GulfFlow products, is outside the scope of this paper on the dataset generation itself. However this brief comparison illustrates that the GulfFlow products are largely in agreement with, but are a substantial improvement on, other data products for the region.

## 3 Data sources

This section presents in detail the properties of the various drifter datasets from the Gulf of Mexico that are aggregated here.
Drifter data in the Gulf are available from 15 different sources, presented in Fig. 4 and listed in Table 1. The figures and table represent the state of the datasets after the uniform processing methodology discussed subsequently in Sect. 4. Instructions for obtaining the various datasets may be found Sect. 7, "Code and data availability", near the end of the paper. The data sources are now described in chronological order.

### 3.1 The Texas-Louisiana Shelf Circulation and Transport Study (LATEX)

The Texas-Louisiana Shelf Circulation and Transport Study (LATEX) was an early Lagrangian experiment to study the circulation on the Texas-Louisiana Shelf. LATEX-A consisted of nineteen drifters released between August 1992 and November





| | | | | | | | | | | | |
|---|---|---|---|---|---|---|---|---|---|---|---|
| Name | Type | Drogue | Tracking | Δ | # Traj | # Points | % Fill | First Date | Last Date | Duration | Max |
| LATEX | WOCE | 7.5* m | Argos | 6.0 | 17 | 33792 | 2.201 | 03/08/92 | 19/02/95 | 83 ± 73 | 251 |
| SCULP1 | CODE | 1 m | Argos | 1.5 | 378 | 570163 | 0.196 | 02/06/93 | 29/01/95 | 63 ± 39 | 131 |
| SCULP2 | CODE | 1 m | Argos | 1.5 | 247 | 387946 | 0.555 | 06/02/96 | 31/10/96 | 65 ± 41 | 224 |
| GDP | SVP | 15 m | Argos | 6.0 | 73 | 105703 | 0.043 | 25/09/96 | 01/07/19 | 60 ± 82 | 403 |
| HARGOS | SVP | 15 m | Argos | 1.0 | 193 | 363311 | 2.081 | 20/01/99 | 22/04/17 | 78 ± 94 | 593 |
| AOML | CODE | 1 m | Argos | Irreg. | 76 | 76314 | 2.029 | 10/12/03 | 30/05/12 | 42 ± 25 | 95 |
| SGOM | FHD | 45 m | GPS | 1.0 | 458 | 510139 | 0.132 | 25/09/07 | 21/09/14 | 46 ± 47 | 254 |
| NGOM | FHD | 45 m | GPS | 1.0 | 370 | 461516 | 4.444 | 15/02/10 | 02/09/14 | 52 ± 48 | 273 |
| OCG | CODE | 1 m | Argos | 0.5/1.0 | 59 | 51212 | 0.499 | 30/04/10 | 29/01/13 | 36 ± 24 | 99 |
| GLAD | CODE | 1 m | GPS | 0.25 | 297 | 391442 | 0.004 | 20/07/12 | 22/10/12 | 55 ± 29 | 94 |
| Hercules | Tube | 1 m | GPS | 5 min | 12 | 9322 | 3.100 | 27/07/13 | 10/09/13 | 32 ± 10 | 45 |
| HGPS | SVP | 15 m | GPS | 1.0 | 39 | 128090 | 0.169 | 07/08/13 | 31/03/19 | 137 ± 140 | 673 |
| LASER | CARTHE | 1 m | GPS | 0.25 | 996 | 891174 | 0.109 | 20/01/16 | 30/04/16 | 37 ± 18 | 89 |
| DWDE | Various | 1 m | GPS | 1.5 | 207 | 410833 | 0.575 | 21/06/16 | 18/04/18 | 83 ± 58 | 294 |
| SPLASH | CARTHE | 1 m | GPS | 5 min | 339 | 101487 | 5.774 | 19/04/17 | 08/06/17 | 12 ± 11 | 48 |
| GD_All | Various | Various | Various | 1.0 | 3761 | 4492444 | 0.987 | 03/08/92 | 01/07/19 | 50 ± 48 | 673 |
| GD_Open | Various | Various | Various | 1.0 | 2726 | 3109956 | 0.668 | 03/08/92 | 01/07/19 | 48 ± 47 | 673 |

**Table 1.** Meta-information for the various surface drifter datasets in the Gulf of Mexico. From left to right, the columns are: experiment name; drifter type; nominal drogue depth; tracking system; nominal original sample interval Δ in hours; number of different trajectory segments; number of hourly data points after interpolation; percent of these that qualify as "filled" as described in the text; date of first hourly data point in DD/MM/YY format; date of last hourly data point; mean duration of trajectories after processing, in days, plus or minus the standard deviation of trajectory durations; and maximum trajectory duration in days. Different experiments are sorted in order of the date of the first data point appearing in the processed dataset. For the nominal drogue depth, the approximate extension of the drogue below the water surface is used for the CODE and CARTHE drifters, the mid-depth of the holey sock drogue is used for the SVP and WOCE drifters, and the nominal parachute depth is used for the FHD drifters. The DWDE experiment contains three different types of drifters, CODE, Microstar, and Doris drifters, described further in the text. *The LATEX drifters had a 7.5 m drogue depth, apart from three drifters, see Sect. 3.1. The last two lines refer to the GulfDriftersAll dataset created herein this is the basis for the GulfFlow gridded product, and the GulfDriftersOpen version containing only publicly-available data.



**Figure 4.** Surface drifter data in the Gulf of Mexico from fifteen different sources, as labeled, together with the combined dataset in the last panel. Colored lines are different trajectories from the processed dataset, the beginning of each of which is marked by a black dot.





**Figure 4.** Continued from previous page.



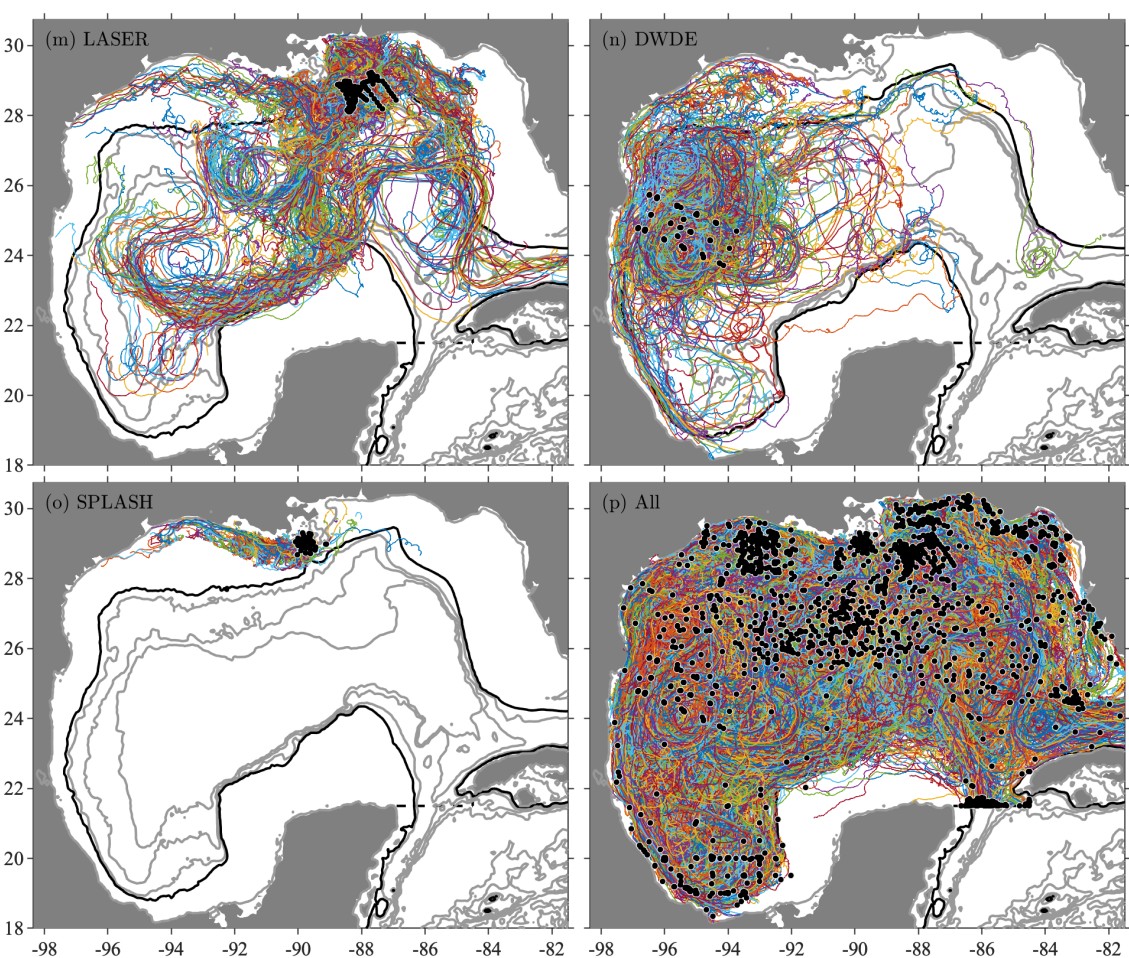

**Figure 4.** Continued from previous page.

1994, see Fig. 4a; data from a related experiment, LATEX-C, could not be located. The drifters, referred to as "WOCE-type" drifters by Howard and DiMarco (1998), are described by those authors as follows:

> The drifters consisted of a spherical, 33.7-cm diameter, foam-filled fiberglass surface float attached by a tether to a 91-cm diameter hoop which supported a 6-m cylindrical drogue made of heavy canvas. The canvas cylinder had a series of circular holes in it, which is why this type of drogue is commonly referred to as a "holey-sock". Eighteen drifters had a 3-m tether which placed the bottom of the 6-m drogue at 9-m depth. Two drifters (07834 and 07833) had longer tethers which placed the drogue bottom at 50-m depth and one (07839) had an even longer tether which placed the drogue bottom at 100-m depth.





Thus the nominal drogue depth of most drifters was at 7.5 m. For this data, raw position estimates from Argos tracking had been spline-fitted onto six-hourly trajectories. In our processing, brief initial deployments and recoveries of drifters 69341 and 78331, lasting only a few days, are omitted.

### 3.2 The Surface Current and Lagrangian Drift Program (SCULP)

The Surface Current and Lagrangian Drift (SCULP) Program described by Ohlmann and Niiler (2005) consisted of three sep-
arate experiments: SCULP-I, which focused on the Texas-Louisiana Shelf beginning in June 1993; SCULP-II, which focused on the West Florida Shelf beginning in February 1996; and SCULP-III, which sampled eddies in the Gulf beginning in April 1998. Of these, only the first two are available, denoted SCULP1 and SCULP2 here. The trajectories are shown in Fig. 4b,c and are seen to provide dense coverage over much of the U.S. continental shelf in the Gulf.

The SCULP experiments used Argos-tracked drifters patterned after the Coastal Dynamics Experiment (CODE) drifters of
Davis (1985) and manufactured by Technocean, Inc., http://www.technocean.com. These drifters have submerged sails roughly 1 m wide by 1 m deep acting as a simple drogue, and take the form of a plus sign (+) when viewed from above.

Upstream processing of the SCULP datasets is described by Ohlmann and Niiler (2005), and included despiking by flagging time points where velocities exceeded 250 cm s$^{-1}$. A final interpolation step is described as follows:

> The despiked position data were then interpolated onto a uniform three-hour time grid by fitting an analytic corre-
> lation function to the Fourier transform of a model spectrum based on 10 days of unequally spaced data centered
> on the day of interest (Ohlmann et al., 2001; Van Meurs, 1995). The correlation function includes parameters to
> represent a low-frequency spectral amplitude, a tidal amplitude, and a tidal peak width.

This represents a very different and more complex interpolation step than has been employed in any of the other drifter datasets, and calls for specialized processing steps to detect occasional oscillatory artifacts that are discussed later. While it appears that
such events are rare, and although we have done our best to identify suspect time periods, the possibility of such artifacts should be kept in mind.

### 3.3 NOAA's Global Drifter Program (GDP) drifters

The United States' National Oceanic and Atmospheric Administration (NOAA) produces a large global dataset of surface drifters through its Global Drifter Program (GDP), http://www.aoml.noaa.gov/phod/dac. The physical design of the GDP
drifters is based on instruments developed for the Surface Velocity Program (SVP) of the Tropical Ocean Global Atmosphere (TOGA) experiments, see Lumpkin and Pazos (2007). Consequently, the drifters employed by the GDP are known as "SVP drifters". While there are several design variants, a common feature is a holey-sock drogue centered at 15 m depth that is intended to reduce wind slippage. Drogues can be lost during the drifter lifetime, which will alter the response to wind forcing. A flag for the presence or absence of the drogues is provided as a part of the GDP dataset using the methods described by
Lumpkin et al. (2013).





The standard GDP dataset is a 6-hourly product that uses the quality control process of Hansen and Poulain (1996), as well as the kriging method of interpolation described therein. This processing involves heavy interpolation, dating back to a time when the typical temporal density of position fixes was much less than it is currently. Position fixes were historically determined using Argos tracking, but since 2013 a steadily increasing fraction of drifters has been tracked with the Global Positioning System (GPS). Further details on this dataset may be found in Lumpkin and Pazos (2007).

An updated, higher temporal resolution version of this dataset is the hourly product of Elipot et al. (2016), constructed using local polynomial fitting or "loess" (Fan and Gijbels, 1996; Cleveland, 1979). While the hourly dataset mostly contains data after 2005—when a change to the tracking arrangement with Argos led to many more position fixes—it also contains some trajectories at earlier times when the average sampling rate happened to be sufficiently high. Drifters tracked by the Argos system, as well as those using the much higher accuracy GPS tracking, are both included; see Elipot et al. (2016) for a detailed discussion of the errors expected for each of these two tracking methods.

Because of the very different tracking and interpolation methods employed, the GDP drifter dataset is separated into three distinct portions: hourly Argos (HARGOS) and hourly GPS (HGPS) trajectories from Elipot et al. (2016), and trajectories from the standard product (GDP). Trajectories that are also in either HARGOS or HGPS are omitted from the GDP portion to prevent redundancy. Plots of these trajectories are shown in panels (d), (e) and (l) of Fig. 4. As these drifters are generally launched outside of the Gulf of Mexico, one sees them often entering via the Yucatán Channel. For the HARGOS dataset one also sees many starting points in the interior of the Gulf, due either to local deployments or because these mark the starting points of trajectory segments with sufficiently dense sampling to be included in the hourly product. A tendency to typically not cross from deep water to the shallow waters of the continental shelf is apparent.

### 3.4 AOML South Florida Program and Hurricane Response Drifters

Drifter trajectories from two small experiments by AOML, the South Florida Program (SFP) and Hurricane Response Drifters (HRD), both using CODE-type drifters tracked by Argos, are shown in Fig. 4f. These drifters do not appear to have been used in a previous scientific publication. These will be grouped together under the category "AOML". Most of these were deployed at irregular intervals between 2003 and 2012 off the west coast of Florida, although some were deployed during 2005 on the Texas-Louisiana shelf. Unlike all the other datasets used here, these data are distributed in the form of raw position fixes, and therefore require an additional processing step. The raw position fixes are bin-averaged onto a uniform hourly grid, then gaps up to six hours are filled with interpolation using a piecewise cubic Hermite interpolation polynomial, or the "pchip" method.

### 3.5 Southern and Northern Gulf of Mexico experiments (SGOM & NGOM)

Two datasets analyzed in this project use Far Horizon Drifters (FHD) manufactured by Horizon Marine, now a part of the Woods Hole Group (http://woodsholegroup.com), consisting of a cylindrical surface buoy attached to a 45 m line terminating in a 1.2 m parachute-style drogue. These instruments are deployed by air, during which process the drogue doubles as an actual parachute, and record their positions at hourly intervals using GPS. The Far Horizon Drifters are discussed in Anderson and Sharma (2008) and Sharma et al. (2010). Unlike the SVP-type drifters, there is no automated mechanism for detecting





drogue presence, nor study of drogue presence as far as we are aware; thus one should be aware of the potential impact of wind
slippage, discussed further subsequently.

The first dataset of this type is from the Southern Gulf of Mexico (SGOM) drifters. An earlier version of this dataset
was previously utilized in a study by Pérez-Brunius et al. (2013). According to those authors, three to five drifters were air-
deployed every month in the Bay of Campeche south of 20.5° N beginning in October 2007; this deployment continued through
mid-2014. A second set of Far Horizon Drifters is the Northern Gulf of Mexico (NGOM) dataset seen in Fig. 4h, largely
contemporaneous with the SGOM dataset but deployed in the U.S. waters. An important point is that the NGOM drifters
were preferentially deployed in order to sample eddies as a part of Horizon Marine's EddyWatch program, and therefore do
not represent an independent and unbiased sampling of the circulation. In the upstream processing of both of these datasets,
position fixes were linearly interpolated between gaps, and data points were then flagged as bad if position fixes were located
on land, if speeds exceeded 300 cm s$^{-1}$, and during gaps of larger than six hours.

## 3.6 Ocean Circulation Group (OCG) drifters

A relatively small set of drifters is available from the Ocean Circulation Group (OCG) at the University of South Florida.
These data are from two separate experiments, the OilSpill experiment in the immediate aftermath of the Deepwater Horizon
oil spill in April 2010, with deployments on the West Florida Shelf, and a very small coastal experiment in 2012 called
RedTide deployed on the Texas-Louisiana Shelf. Drifters from both experiments are grouped together and shown in Fig. 4i.
The OilSpill drifters were analyzed in Liu and Weisberg (2011) and Liu et al. (2013b). These and the RedTide drifters are
CODE-type drifters manufactured by Technocean (Y. Liu, pers. comm.) and tracked by Argos (J. Donovan, pers. comm.). The
temporal resolution was hourly, with occasional gaps, for most of the drifters, with a subset of 36 OilSpill drifters having
half-hourly resolution.

## 3.7 The Grand Lagrangian Deployment (GLAD)

The Grand Lagrangian Deployment (GLAD) (Özgökmen, 2013; Poje et al., 2014) was a major experiment designed to examine
dispersion in the aftermath of the Deepwater Horizon oil spill. This experiment was carried out by the Consortium for Advanced
Research on Transport of Hydrocarbon in the Environment (CARTHE). The GLAD experiment utilized ∼300 CODE-type
drifters with GPS tracking. Trajectories are shown in Fig. 4j. Distinguishing features of this experiment are that the drifters
were launched within three weeks of each other and were grouped into triplets, separated by about 100 m, in order to study
small-scale dispersion.

Detailed information as to the data processing is distributed with the data. Position fixes were obtained roughly every five
minutes. Data points were then flagged as bad if velocities exceeded 300 cm s$^{-1}$ or met several other quality-check criteria.
Valid positions were then spline-interpolated to uniform 5-minute time intervals, filtered with a one-hour low-pass filter, and
finally interpolated onto a 15 minute temporal grid. Drifter records end when the drifter was determined to have been picked
up by a boat, when the signal was lost for more than 24 hours, or when the drifter displacement exceeded 80 km in a 12-hour
period.





Since the interest in this experiment was on short-timescale dispersion, the drifters were not tracked for a particularly long period of time. Drifter records end abruptly on October 22, 2012, with no trajectories longer than 95 days, see Table 1 and also the subsequent Fig. 5. Therefore, this experiment represents an intensive sampling over a short time.

### 3.8 The Hercules experiment

Hercules was a relatively small experiment with nineteen drifters launched near the site of the Hercules 265 drilling rig in July 2013, and intended to track dispersion in the aftermath of an explosion on that rig (Özgökmen, 2014; Weber et al., 2016), show in Fig. 4k. The drifters were tracked with GPS with positions reported every five minutes. The drifter designs were of two different experimental types, thirteen of type "A" and six of type "B". However, visual inspection shows that the trajectories from type B drifters are apparently poorly sampled and also of short duration, and consequently these six are discarded. The type A drifters are described in the dataset documentation as having a "plastic, tubular body roughly 50 cm high" and will be denoted "Tube"-type drifters. These may be considered as related to the CODE drifters in that they are drogued close to the surface.

### 3.9 The Lagrangian Submesoscale Experiment (LASER)

The goal of the recent Lagrangian Submesoscale Experiment (LASER), also carried out by CARTHE, was to examine dispersion by submesoscale processes in wintertime conditions in the northeastern Gulf of Mexico (D'Asaro et al., 2017; Haza et al., 2018). For this experiment, an innovative new type of drifter—the CARTHE drifter—was designed that is inexpensive, mostly biodegradable, and easy to deploy in large numbers (Novelli et al., 2017; Lumpkin et al., 2017). It consists of a GPS-tracked toroidal float connected to a plus-shaped drogue that extends about 60 cm below the surface. Laboratory experiments (Novelli et al., 2017) showed that the drifting characteristics of the CARTHE drifters are essentially identical to those of the earlier CODE drifters.

Over 1000 CARTHE drifters were deployed in January and February 2016 in the northeastern Gulf, in the vicinity of DeSoto Canyon, see Fig. 4m. These were deployed in three sets of more than 300 drifters each, again with many of the drifters deployed in triplets in order to study dispersion, and rapidly spread throughout the Gulf. Like GLAD, this experiment represents a very intensive sampling over a short time, with no trajectories longer than 90 days. It was found during this experiment that the CARTHE drifters occasionally lose their drogues, so consequently a drogue presence flag was determined by Haza et al. (2018) by analyzing both trajectory response and transmission information, and distributed as Haza et al. (2017). The drifter design was later improved to help prevent this problem in the future (Novelli et al., 2017).

### 3.10 The Deep Water Dispersion Experiment (DWDE)

The Deep Water Dispersion Experiment (DWDE) was designed to study dispersion in the deep western Gulf of Mexico. This experiment was carried out by CICESE in four separate deployments, with a total of 207 drifters: June 21–24, 2016 (45 drifters); October 15–19, 2016 (55 drifters); April 25–29, 2017 (56 drifters); and November 7–10, 2017 (51 drifters). This experimental





design allowed the surface velocity field to be sampled with relatively high spatial resolution in two different seasons and in two different years. DWDE is available only for noncommercial use, see Sect. 7, and as such it cannot be included as a part of
our freely distributed drifter dataset, though investigators can access it separately.

The DWDE experiment used drifters of three different designs, all tracked by GPS and all with a 1 m drogue depth. In the first two deployments, most were Microstar drifters as used in e.g. Ohlmann and White (2005), which register a flag if their drogue is lost. In the second two deployments, most drifters were of the CODE type described above. During both years, a small number of drifters were of the "Doris" type, a simple drifter manufactured by the Observatorio Oceanográfico Regional Costero
group from Universidad Nacional Autónoma de Baja California, México (https://oorco.ens.uabc.mx/sondas-oceanograficas). Upstream processing was the same as for the SGOM experiment.

### 3.11 Submesoscale Processes and Lagrangian Analysis on the Shelf (SPLASH)

The Submesoscale Processes and Lagrangian Analysis on the Shelf, or SPLASH, experiment was designed to study nearshore dispersion in the Louisiana Bight in the spring of 2017 (Huntley et al., 2017). More than 300 GPS-tracked CARTHE-type
drifters were released. The dataset used here has been pchip-interpreted to five minute intervals after removing points with velocities exceeding 262 cm s$^{-1}$ or accelerations above 1.0 cm s$^{-2}$. This dataset does not appear to have yielded scientific publications at the time of this writing.

### 3.12 Other datasets

Apart from the previously discussed proprietary Horizon Marine drifters used in Mulet et al. (2020), that are not freely available
and to which we only have access for that the portion in the NGOM experiment, the above datasets represent nearly all remotely tracked surface drifter experiments conducted in the Gulf of Mexico that are referred to in the peer-reviewed literature. Nowlin et al. (2001) mentions another small early experiment conducted during the 1990's, NEGOM, that we have been unable to locate. This, along with the LATEX C and SCULP-III experiments, mentioned above, appear to have been lost.

## 4   GulfDrifters, a consolidated drifter dataset

All datasets described in the previous section are subjected to a uniform processing methodology. The result is a quality-controlled dataset, called GulfDrifters, that has been interpolated onto an hourly time grid, with time points corresponding to gaps filled during our interpolation flagged as such. The processing steps are described in this section, followed a discussion of bias and error sources, and finally a presentation of the sampling properties of GulfDrifters itself. GulfDrifters is created in two versions, GulfDriftersAll that is the basis for GulfFlow, and GulfDriftersOpen that contains only publicly available data
(excluding NGOM, SGOM and DWDE) and that is freely distributed as described in Sect. 7.





## 4.1  Uniform processing

The uniform processing methodology is as follows. We begin with data that has been interpolated to a uniform sampling interval, possibly with gaps. Trajectory segments lying entirely within the study region shown in Fig. 1 are isolated. Depth is found by looking up drifter positions within the Smith and Sandwell global one minute bathymetry dataset v. 19.1 (Sandwell

and Smith, 1997). Data points for which the depth is negative, indicating a location on land, are flagged as bad.

A visual inspection is then carried out in order to identify trajectory segments that appear suspicious. Several different types of features are interpreted as cause to flag a data segment: stationary locations, likely indicating a grounded drifter; extended periods of linearly varying positions, likely indicating a linear interpolation over a data gap; isolated, patchy data segments of valid data near the end of a record; unusually noisy or jagged data segments; high-speed segments terminating

near the shore, likely arising from grounding due to wind or wave activity; isolated anomalous points; and finally conspicuous, rapidly-changing oscillations. These latter only in the SCULP drifters and apparently indicate a gap that has been filled with the vigorous interpolation applied to those datasets. Such features are clearly distinguished from eddies in that they appear as "knots" rather than loops when viewed on a map.

All data points that have been flagged as bad or missing are removed. The data are then interpolated to a uniform hourly

spacing using interpolation with a piecewise cubic Hermite interpolation polynomial, the "pchip" method. A `filled` flag is created to indicate when this interpolation has been applied. Within the hourly dataset, the `filled` value of interpolated data point is set to true if no valid un-interpolated data points were present within plus or minus six hours and five minutes. In the case of the hourly HARGOS and HGDP datasets, a `gap` field is available that gives the time gap that has been interpolated over in the upstream interpolation. For these two datasets, our `filled` flag is also set to true if the time gap in the upstream interpolation

exceeds six hours. For the other datasets, similar information regarding upstream interpolation gaps is not available.

From this interpolated hourly dataset, velocity is computed using the first central difference on the sphere, see Appendix A. Trajectory beginning or ending segments lacking good data are discarded, as are any trajectories containing no good data. The remaining trajectories from all experiments are combined, with an integer `source` field added to indicate the originating dataset.

Next, acting on the combined hourly dataset, several objective criteria are applied to identify possibly problematic points. Data points having instantaneous speeds less than 0.1 cm s$^{-1}$ or greater than 250 cm s$^{-1}$ are flagged, as well as those failing to pass a minimum acceleration criterion. The minimum acceleration criterion identifies times for which the acceleration magnitude $|u'(t) + \mathrm{i}v'(t)|$, smoothed with a 168-point (two week) Hanning filter, is smaller than $10^{-4}$ cm s$^{-2}$; time periods exhibiting such a high degree of smoothness generally either reflect that the drifter is grounded, or that a data gap that has been

interpolated over in earlier processing. Two criteria are also applied to isolate several unrealistically fast segments in shallow water seen in the SCULP datasets.

Data flagged at this stage is marked by setting the `filled` field to true so that they can be excluded from future analysis, and then the flagged values are re-interpolating over using pchip. The fraction of data points that are filled at this secondary level is very small, about two per thousand valid data points. Finally, velocity and depth are both re-computed in the same





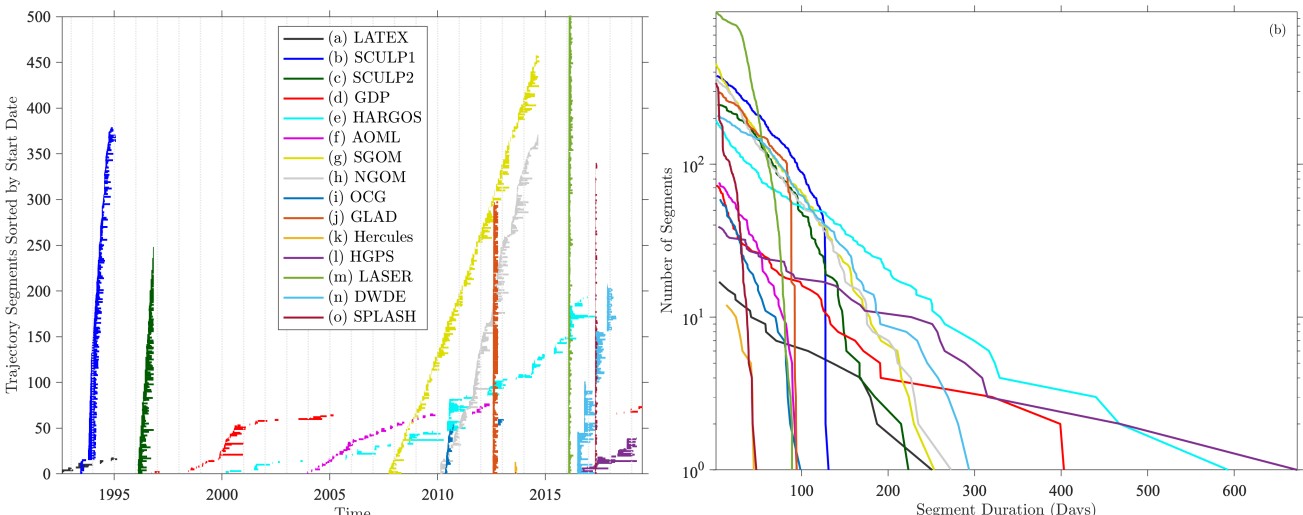

**Figure 5.** Temporal ranges (a) and durations (b) of drifter trajectory segments from the various sources, after the processing described in Sect. 4.1. In panel (a), each small horizontal line marks the temporal range of a different drifter trajectory. The $y$-axis is the trajectory segment number within each data source, sorted in order of the date of the first measurement point. For display purposes, the upper limit of the $y$-axis is set to 500 trajectories, although the LASER experiment has nearly 1000 trajectories. In panel (b), the lines show the number of trajectories exceeding a specified duration with a logarithmic $y$-axis. The lower limit of the $y$-axis is one, and gives the duration of the longest trajectory segment in each dataset.

manner as described above. Processing of the SCULP data is double-checked by computing the deviation of the speed from its median value within grid boxes over all drifters, and noting no readily evident difference between the SCULP and non-SCULP speed values.

The result of these processing steps is our consolidated product, GulfDriftersAll, summarized in the second-to-last line of Table 1 and presented in Fig. 4p. A second version, GulfDriftersOpen, which excludes the NGOM, SGOM, and DWDE

datasets for reasons discussed earlier, is summarized in the last line of Table 1 and is distributed to the community without restriction as discussed in Section 7. Note that DWDE is separately available for noncommercial use.

### 4.2 Bias and error considerations

The GulfDrifters dataset is quite heterogeneous, reflecting different drifter designs, drogue depths, and tracking methods, as well as variation in the upstream interpolation and processing steps. These differences, as well as other sources of potential

bias or error, will now be discussed in more detail. For references, the temporal extent of the various component experiments is presented in Fig. 5a, while distributions of the trajectory durations are shown in Fig. 5b.

The most obvious distinction between the experiments is drifter design. Due to the different drogue depths listed in Table 1, the various experiments are tracking the currents at different depths. Moreover, there is the issue of possible drogue loss. The CODE-type drifters are unlikely to transmit in the absence of a drogue due to their construction. For the others, drogue loss is a



concern as it impacts the wind slip. For the SVP-type drifters, the impact of drogue loss is to increase the wind slip from 0.1%
of the 10-m wind speed to 0.7–1.6%, see Laurindo et al. (2017) and references therein. For the CARTHE drifters, Novelli et al.
(2017) report that drogue loss increase the wind slip form less than 0.5% of the 10-m wind speed to as much as 2%. We are
unaware of published results for the Far Horizon drifters, but would expect the results to be similar.

Drogue presence flags are available for all the SVP-type drifters (GDP, HARGOS, and GPS), see Lumpkin et al. (2013), as
well as for the LASER experiment using the CARTHE drifters (Haza et al., 2018) and the Microstar drifters used in about half of
the DWDE experiment. Such a flag is not currently available for the SCULP, OCG, Doris, or Far Horizon (SGOM & NGOM)
drifters. One could be created by examining the response of the drifters to wind forcing, following Lumpkin et al. (2013)
and Haza et al. (2018), however, this would be a substantial undertaking that is outside the scope of this paper. Examining
correlations of different types of drifters with winds and with currents from CMEMS gridded products in a study of the Bay
of Campeche circulation, Pérez-Brunius et al. (2018) conclude that "the FHD drifter data have the same correlation with the
winds as the 15-m drogued SVP drifters from the Poulain et al. (2002) study, and are highly correlated with the geostrophic
currents derived from altimetry. Both results show that the FHD drifter data represent well the meso- and large-scale features
of the velocity field in the upper layer of the Bay of Campeche." This indicates that drogue loss from the Far Horizon drifters
is unlikely to present a significant problem.

The various experiments differ in their temporal distributions, another factor that can affect the ways that these experiments
sample the circulation. The temporal distributions are clearly seen in Fig. 5a. Whereas some experiments (e.g. GLAD, Hercules,
LASER, and SPLASH) involved sudden deployments and also sudden terminations, lasting only a few months, others (e.g.
SGOM, NGOM, and DWDE) involved deployments over a long period of time. The GDP group of drifters—GPD, HARGOS,
and HGPS—are different from the others in that they generally enter the Gulf by chance, also leading to temporal distributions
that are spread out in time. As mentioned above, NGOM is unique in that it was from a program to monitor eddies, and
therefore may contain a bias toward a state of eddy presence.

A less obvious distinction between experiments is the difference in trajectory durations. As seen in Fig. 5b, durations for
the various experiments fall into three groups that appear to reflect drifter design. The longest-duration trajectories are all
associated with SVP-type drifters in the GPD, HARGOS, and HGPS datasets, each of which have at least one trajectory
exceeding 400 days. Moreover, these duration curves have a shallower slope than those for the other experiments, indicating
that the SVP drifters are more likely to experience long lifetimes. Among the shortest-lifetime experiments are GLAD, LASER,
and SPLASH, with their sudden cut-off times, none of which have trajectories exceeding 100 days. One also sees the CARTHE-
type drifters (LASER and SPLASH) have a steeper slope than other experiments, indicating a higher failure rate. These duration
differences mean that the ability to resolve the low-frequency behavior also differs among the experiments.

Another important issue is that of position accuracy. Whereas the Argos tracking system has typical positioning errors of
hundreds of meters, see Tables 1 and 2 of Elipot et al. (2016), GPS positions are accurate to within a few meters. A very
detailed treatment of the errors associated with Argos positioning can be found in Sect. 2.3 of Elipot et al. (2016), so we refer
the reader there for further details. A practical impact of these tracking differences are that GPS-tracked drifters have almost no
bad data points. While both Argos- and GPS-tracked drifters can be productively used to study the flow on monthly or longer



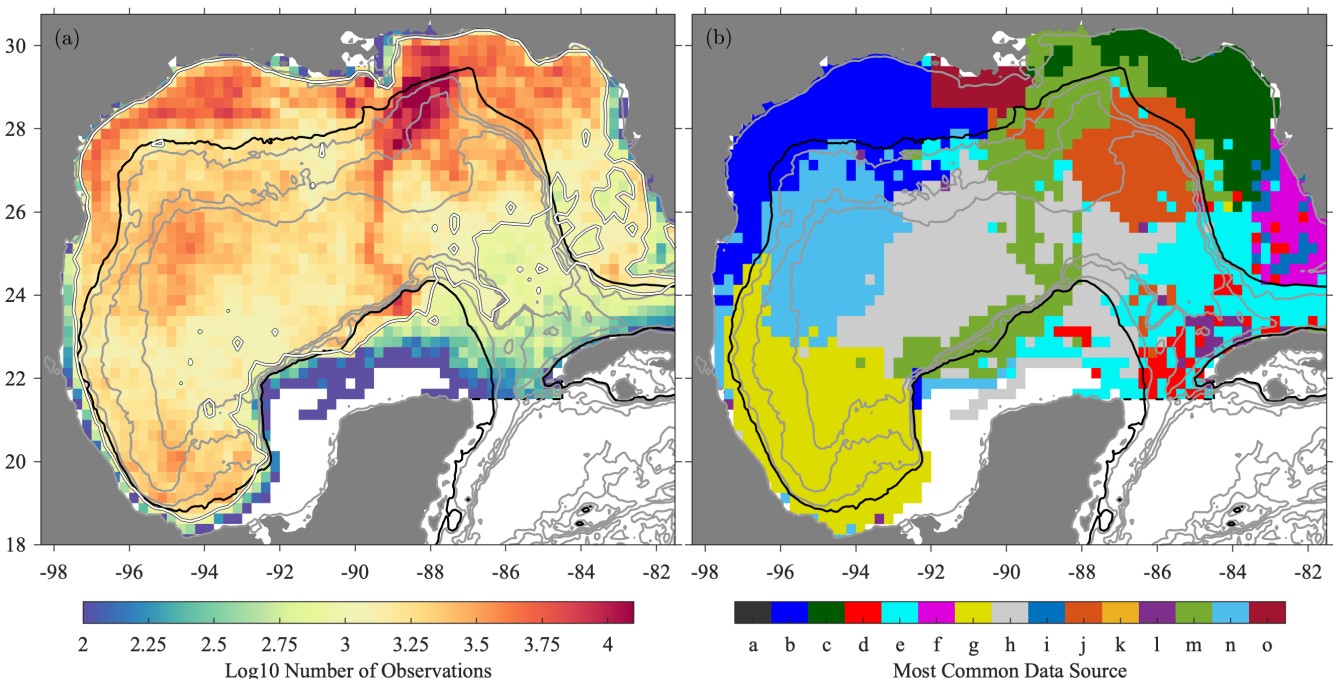

**Figure 6.** The spatial distribution of temporal sampling characteristics: (a) the number of hourly observations in quarter-degree bins presented on a logarithmic scale, together with (b) the most common data source within each bin. The heavy white contour in (a) is $10^3$ observations. The key to the letter codes in (b) is given in Fig. 5.

time scales, the GPS drifters can resolve fast timescale, small-amplitude signals—such as internal waves or small-scale vortex motions—that are well below the noise level of the Argos-tracked drifters.

### 4.3   Sampling properties

The spatial distribution of hourly, non-filled observations from GulfDriftersAll in quarter-degree bins is shown in Fig. 6, together with the most commonly-occurring sample source within those bins. A highly inhomogeneous sampling is seen. The

various experiments are spatially complementary, with different experiments dominating in different regions.

The Gulf is bisected north-south by a ridge of very high sample densities, seen to be associated with the LASER experiment. High densities are also seen on the Texas-Louisiana Shelf and West Florida Shelf, respectively associated with SCULP1 and SCULP2 together with LASER. Moderately high densities in the western central and southern Gulf are associated with DWDE and SGOM respectively. The Mississippi outflow region is most commonly sampled by the SPLASH experiment. Low data

densities are seen in the southeastern Gulf, coincident with the southern three-quarters or so of the Loop Current. There, deployments elsewhere within the Gulf of Mexico tend not to reach, and the dominant sampling is therefore associated with inflowing GDP, HARGOS, and HGD drifters. Very low or zero densities are observed along most of the Yucatán shelf.





An implication of this inhomogeneous sampling pattern is that the currents observed by the consolidated drifter dataset reflect the flow at somewhat different depths, and also different time periods, in the various regions.

## 4.4 A naïve mean flow map

The most obvious way to form an estimated mean flow from the drifters is simply to take the average of all available velocities within each spatial bin. The result, presented earlier in Fig. 2c, is seen to have a distorted and unrealistic appearance in the vicinity of the Loop Current. The reason for this can be understood at once by looking at the data distribution in Fig. 6. The ridge of high southward velocities to the west of the Loop Current is coincident with the region of extremely dense sampling due to the LASER experiment.

This is a simple yet important message. When drifter data is distributed highly inhomogeneously in time, one does not wish to simply average it. Such an average tends to bias the result towards the state of the system at the times of densest observations. Fortunately, a small modification in the temporal averaging will lead to a substantial improvement.

## 5 GulfFlow, 3D gridded velocity products

Two factors motivate the creation of gridded velocity products for the Gulf of Mexico derived from surface drifters. The first is a desire to study the mean circulation, along with seasonal and interannual variability, in a way that avoids the averaging artifacts just discussed. The second is the aspiration to make information derived from the consolidated drifter dataset available to the community, even if some of the trajectories themselves cannot be distributed.

This section describes the creation of the gridded products, examines its sampling distributions, and uses it to create the improved maps presented in Sect. 2. Errors relative to the unknown true mean, and the improvement over the naïve map of the previous section, are quantitatively estimated.

### 5.1 Creation of the gridded product

A gridded product, called GulfFlow is created by averaging all available data from the GulfDriftersAll dataset within spatial bins, and within overlapping month-long temporal bins having a semimonthly spacing. Two versions are created, GulfFlow-$1/4°$ that uses quarter-degree spatial bins, and GulfFlow-$1/12°$ using one twelfth-degree spatial bins. The dataset spans August 16, 1992 to August 1, 2019, for a total of 648 overlapping time slices. Odd-numbered slices correspond to calendar months, while even-numbered slices run from halfway through one month to halfway through the following month. In addition to the average velocities within each 3D bin, the count of sources contributing to each bin is also distributed, as is the subgridscale velocity variance discussed in the next section.

The `count` variable is a four-dimensional array of integers, the fourth dimension of which has length 30. This variable gives the number of hourly observations from each source dataset contributing to each three-dimensional bin. Values 1–15 are the count of velocities from drifters from each of the 15 experiments that have not been flagged as having lost their drogues, while



values 16–30 are for observation from drifters that have been flagged as having lost their drogue. Values above 15 are only populated for the GDP, HARGOS, LASER and some of the DWDE drifters, as a drogue presence flag is not always available.

It is useful at this stage to introduce notation for different types of averages. For convenience we represent the velocity as a vector, $\mathbf{u} \equiv [u \ \ v]^T$, where the superscript "$T$" denotes the transpose. Let an overbar, $\overline{\mathbf{u}}$, denote an average over a spatial bin and over all times, and let angled brackets, $\langle \mathbf{u} \rangle$, denote an average over a spatial bin and a particular temporal bin. Thus, $\langle \mathbf{u} \rangle$ is a function of time while $\overline{\mathbf{u}}$ is not. We refer to $\langle \mathbf{u} \rangle$ as the local average, $\overline{\mathbf{u}}$ as the global average, and $\overline{\langle \mathbf{u} \rangle}$ as the double average. Note that these averages do not commute; $\langle \overline{\mathbf{u}} \rangle$ is the same as $\overline{\mathbf{u}}$ and is not equal to $\overline{\langle \mathbf{u} \rangle}$.

The GulfFlows product contain the local average $\langle \mathbf{u} \rangle$ at grid location and each time slice. As we have seen, a global time average over all velocities in a spatial bin, $\overline{\mathbf{u}}$, is a poor way to form an estimate of the time mean currents, see Fig. 2c. A local average followed by the global average, $\overline{\langle \mathbf{u} \rangle}$, gives the improved mean flow estimate seen in Fig. 2d and discussed earlier in Sect. 2.

## 5.2   A variance decomposition

In the same way that one can different define versions of an space/time averaged flow field, one can similarly define different versions of the velocity covariance matrix. Within each bin, three different time-varying covariance matrices are

$$\mathbf{\Gamma} \equiv \left\langle \left( \mathbf{u} - \overline{\langle \mathbf{u} \rangle} \right) \left( \mathbf{u} - \overline{\langle \mathbf{u} \rangle} \right)^T \right\rangle \tag{1}$$

$$\boldsymbol{\mathcal{E}} \equiv \left\langle \left( \mathbf{u} - \langle \mathbf{u} \rangle \right) \left( \mathbf{u} - \langle \mathbf{u} \rangle \right)^T \right\rangle \tag{2}$$

$$\mathbf{\Sigma} \equiv \left( \langle \mathbf{u} \rangle - \overline{\langle \mathbf{u} \rangle} \right) \left( \langle \mathbf{u} \rangle - \overline{\langle \mathbf{u} \rangle} \right)^T \tag{3}$$

which involve different combinations of local and global averages. The first of these, $\mathbf{\Gamma}$, is the autocovariance within a space/time bin for all observed velocities $\mathbf{u}$ relative to the time-independent double-mean velocity $\overline{\langle \mathbf{u} \rangle}$. The second, $\boldsymbol{\mathcal{E}}$, is the autocovariance within a space/time bin for all observed velocities $\mathbf{u}$ relative to the time-dependent local mean velocity $\langle \mathbf{u} \rangle$ in that bin. The third, $\mathbf{\Sigma}$, is a covariance-like quantity involving the deviation between the local mean velocity $\langle \mathbf{u} \rangle$ in a space-time bin and the double-mean velocity $\overline{\langle \mathbf{u} \rangle}$.

Unlike the other matrices, $\mathbf{\Sigma}$ is not an averaged quantity, but instead is the outer product of a deviation vector with itself; thus it is not technically a covariance. However, its time-average $\overline{\mathbf{\Sigma}}$ will be a covariance matrix, so it is sensible to extend that term to this quantity as well.

The matrices $\mathbf{\Gamma}$, $\boldsymbol{\mathcal{E}}$, and $\mathbf{\Sigma}$ will respectively be called the *total*, *local*, and *bulk* time-dependent covariance matrices. In fact these three matrices are related by the identity

$$\mathbf{\Gamma} = \boldsymbol{\mathcal{E}} + \mathbf{\Sigma} \tag{4}$$

so that the total time-dependent covariance is the sum of the local and bulk covariances. This is shown by substituting

$$\mathbf{u} - \overline{\langle \mathbf{u} \rangle} = \mathbf{u} - \langle \mathbf{u} \rangle + \langle \mathbf{u} \rangle - \overline{\langle \mathbf{u} \rangle} \tag{5}$$





into the definition of $\mathbf{\Gamma}$, followed by carrying out the indicated average. Taking the time average of Eqn. (4) followed by the matrix trace—that is, the sum of the diagonal elements—one finds

$$\gamma^2 = \epsilon^2 + \sigma^2 \tag{6}$$

where $\gamma^2 \equiv \mathrm{tr}\left\{\overline{\mathbf{\Gamma}}\right\}$, $\epsilon^2 \equiv \mathrm{tr}\left\{\overline{\mathcal{E}}\right\}$, and $\sigma^2 \equiv \mathrm{tr}\left\{\overline{\mathbf{\Sigma}}\right\}$, and with "tr" being the trace. This is a partitioning of the velocity variance at each 2D spatial bin into two parts: variability that is resolved by the gridded product, in $\sigma^2$, together with unresolved subgridscale variability, in $\epsilon^2$.

The GulfFlow products contain $\mathcal{E}$ at each 3D bin, in order to quantify the subgrid-scale variability that is lost in forming the local average velocity $\langle \mathbf{u} \rangle$. The bulk covariance $\mathbf{\Sigma}$ is readily formed from the local average velocity $\langle \mathbf{u} \rangle$, and $\mathbf{\Gamma}$, if desired, can be reconstructed from their sum.

The standard deviations $\epsilon$ and $\sigma$ for GulfFlow-$1/4°$ are shown in Fig. 7. The former represents subgrid-scale variability from the original drifter data that is lost in averaging to form the local average velocities $\langle \mathbf{u} \rangle$ on a three-dimensional grid, while the latter represents temporal variability that is resolved by those velocities. Note the difference in magnitudes, as well as the very different spatial patterns. Whereas $\sigma$ is large over the energetic and variable Loop Current, as expected, $\epsilon$ is elevated over the continental shelf in addition to the Loop Current region, and especially in the vicinity of the Mississippi outflow. This difference in spatial patterns is consistent with the physical expectation that variability on the shelf will be dominated by smaller horizontal scale structures than in deep water, see e.g. Bracco et al. (2016) and Luo et al. (2016). Thus, it is likely that meaningful information regarding temporal variability of small-scale energy is contained within the time-varying local variance $\mathcal{E}$.

## 5.3 Sampling properties

The spatial and spatiotemporal sampling of the GulfFlow-$1/4°$ gridded product is shown in Fig. 8. Here, a bin is considered sampled if it contains at least one hourly drifter data point. The spatial distribution map in Fig. 8a shows the percent of possible time slices, out of a total of 648, that are sampled. In general we see that the GulfFlow-$1/4°$ is sparsely sampled, with at most a fifth or a quarter of the overlapping monthly slices being sampled at any spatial bin.

Averaging within monthly slices has led to a distribution that presents much less spatial variability than that seen in the distribution of the original drifter positions in Fig. 6a. Moreover, the nature of the distribution has changed. Total data densities were seen to be generally higher on the Texas-Louisiana continental shelf than in deep water in Fig. 6a. However, the opposite tendency is apparent in Fig. 8a. Here we see that shallow bins are consistently sampled during fewer time slices than deeper bins, with the transition between these roughly coinciding with the 500 m isobath; note the similarity between this isobath and the 10% sampling contour. This in part reflects the fact that the continental shelves have typically been the subject of intense experiments carried out over short durations, as opposed to the repeated deployments over long time periods found in some deepwater experiments. It also reflects a tendency of drifters not to cross from the deep interior to the continental shelves.

One sees that the longitude/time distribution of sampled slices for the GulfFlow-$1/4°$ dataset, Fig. 8b, exhibits a change in behavior between the western and eastern Gulf. Overall the eastern Gulf is considerably less well-sampled than the west.



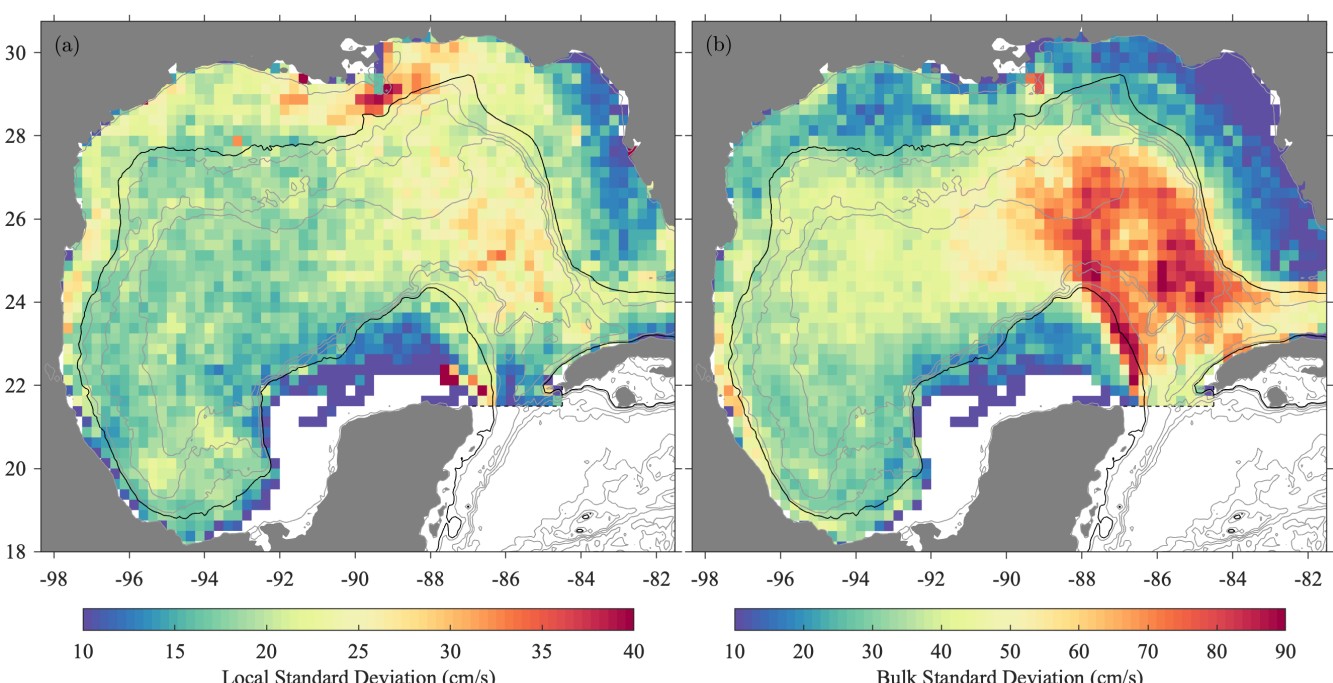

**Figure 7.** The square roots of (a) the local variance $\epsilon^2$ and (b) the bulk variance $\sigma^2$, as defined in the text. Note the different colorbar axes.

Moreover, the late spring and summer months are the least well-sampled in the west, but the best-sampled in the east. These sampling patterns should be kept in mind when examining seasonal or longer time-scale variability.

## 5.4 Assessment of the averaging methods

While the true mean flow in the Gulf of Mexico is unknown, errors in estimating the mean flow from the surface drifter data can
535 still be quantified. This is done by sampling velocities from CMEMS altimetry, as well as the output of high-resolution numerical simulations of the region, at the space/time locations of the observed trajectories. Then applying the straight bin-averaging at one quarter-degree resolution, $\overline{\mathbf{u}}$, or alternatively the double average, $\overline{\langle \mathbf{u} \rangle}$, leads to two different mean flow estimates that can be compared with the "truth" from simply time-averaging the full model or altimetric velocity fields.

Three twenty-year simulations, representing state-of-the-art approximations to the time-varying Gulf of Mexico circulation,
540 are used for this purpose. These are based on three different numerical models: HYCOM, the HYbrid Coordinate Ocean Model, https://www.hycom.org; NEMO, the Nucleus for European Modelling of the Ocean, https://www.nemo-ocean.eu; and ROMS, the Regional Ocean Modeling System https://www.myroms.org. Further details regarding these simulations are not particularly relevant here.

The fields for each of the three numerical models are available over similar time spans, from the beginning of 1992 (or 1993
545 for ROMS) through the end of 2012. Daily averaged fields of surface velocities and sea surface height for each of these three

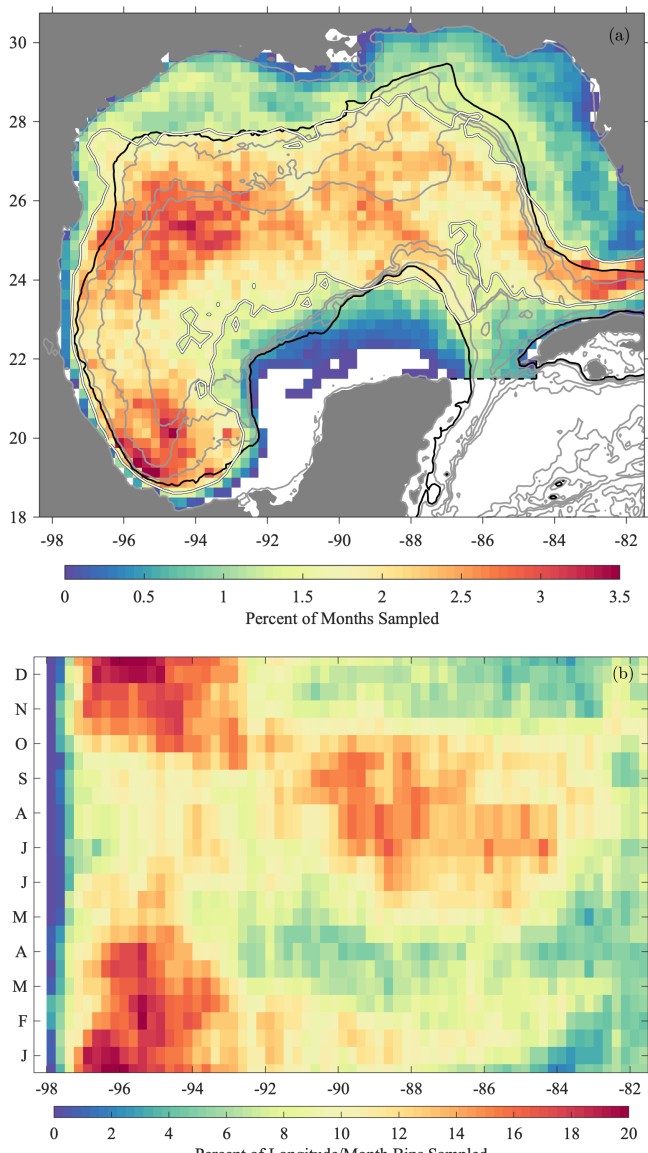

**Figure 8.** Sampling distributions for the GulfFlow-$1/4°$ gridded dataset. The percent of time slices sampled in each quarter-degree spatial bin is shown in (a), while (b) shows the percent of possible longitude/month bins sampled, after summing over all years and all latitudes. The white line in (a) is the 10% sampling contour. In (b), the bins are again $1/4°$ wide in longitude, and are of monthly duration with semimonthly spacing in time.

models are interpolated from the model grid onto a regular $1/40°$ latitude/longitude grid. The CMEMS fields will be left on their original grid, which is the same quarter-degree grid used for GulfFlow-$1/4°$.





Mean Flow Estimation Error Assessment

| Velocity | RMS speed | RMS error I | RMS error II | Reduction |
|---|---|---|---|---|
| CMEMS | 20.1 cm s$^{-1}$ | 13.6 cm s$^{-1}$ | 8.6 cm s$^{-1}$ | 36.9% |
| HYCOM | 29.0 cm s$^{-1}$ | 12.1 cm s$^{-1}$ | 8.2 cm s$^{-1}$ | 32.1% |
| NEMO | 19.0 cm s$^{-1}$ | 10.9 cm s$^{-1}$ | 7.2 cm s$^{-1}$ | 33.6% |
| ROMS | 23.6 cm s$^{-1}$ | 18.5 cm s$^{-1}$ | 10.4 cm s$^{-1}$ | 44.0% |

**Table 2.** An assessment of the errors involved in estimating the time-mean currents from the surface drifter dataset at quarter-degree spatial resolution. This is done using velocities from the time-varying altimetric or model velocity fields interpolated along the observed drifter times and positions, with a modification described in the text. The first column shows the root-mean-square magnitude of the time-mean currents, averaged over all times and over all spatial bins for which the corresponding drifter velocity measurement is defined. The second and third columns are the root-mean-square deviation between the "true" time-averaged currents, and the estimated currents using either method I, the global average $\overline{\mathbf{u}}$, or method II, the double average $\overline{\langle \mathbf{u} \rangle}$. The final column is the error reduction in using method II relative to method I.

The model data is not available after 2012, and as such the model time period does not cover those of the major experiments LASER, DWDE, and SPLASH. Because the goal of this exercise is to determine how the observed spatial patterns of sampling affect velocity reconstructions in the presence of typical variability, is it sensible to fill in this gap after 2012 with model data from an earlier time period. Therefore, in interpolating model velocities onto observed drifter locations, we have subtracted 13 years from sampling times after December 31, 2012; the results are not sensitive to this choice.

The model time-mean flows (not shown) are all broadly similar to the data-derived map of Fig. 2d. The coastal currents and Campeche Gyre are each well captured in two of three models, and the Loop Current behavior is satisfactory in all models. As with the data, estimating the mean flow using the global average $\overline{\mathbf{u}}$ applied to velocities along the observed drifter trajectories lead to artifacts for both CMEMS and the models. Also as with the data, these reconstructions are visually greatly improved using the double average $\overline{\langle \mathbf{u} \rangle}$.

The results are summarized in Table 2. A 32% to 44% percent reduction in root-mean-square error is found using the double average $\overline{\langle \mathbf{u} \rangle}$ rather than the global bin average $\overline{\mathbf{u}}$. Moreover, this table provides a guide to the estimated domain-averaged RMS error associated with the mean flow map seen in Fig. 2d, with a range of 7.2–10.4 cm s$^{-1}$, values that are 28% to 44% of the domain-averaged RMS velocity magnitudes. Thus, while the new map is greatly improved over earlier products, and while the major physical features appear to be resolved, error magnitudes are not negligible relative to the mean current speed, a reflection of the challenges involved in observing the ocean currents.

## 6 Conclusions

The purpose of this work was to gather together all available surface drifter data from the Gulf of Mexico, and to process and distribute it in a form that would be of use to the community. The main product is a space/time gridded products, called GulfFlow, spanning 27 years and distributed at both one-quarter and one-twelfth degree spatial resolution. Source counts—





taking drogue presence into account when possible—as well as subgrid-scale variance are included. Mean flow estimates formed from GulfFlow are shown to be substantially superior to currently available products. It is anticipated that this product will be of value to other investigators working in this environmentally and economically important region.

A second product, GulfDriftersOpen, contains drifter trajectories from all publicly available sources, uniformly processed and quality controlled, and interpolated to hourly resolution. In addition to position and velocity, upstream drogue presence flags are also recorded, as is a flag for bad or missing data encountered during this processing.

A fundamental yet perhaps under-appreciated problem arising when working with drifter data was pointed out, namely that a straightforward bin averaging over all available data can lead to biased estimates of the mean flow. One might think that averaging over more data points leads to improved estimates, but this is only the case if the data points represent statistically independent samples of the field. When many data points are collected in a short time, they will be strongly correlated, and will weight the overall estimate toward the particular state of the field at that time.

Pursuing this further, it is clear that one would like to average over time slices that are comparable in duration to the decorrelation time. A narrower averaging window would lead to bias due to correlated samples, while a longer window would diminish one's ability to reduce variance by subsequently averaging over independent samples. This suggests that further improvement may be obtained by using the altimetric and model velocity fields to estimate the decorrelation timescale, potentially in a spatially-varying context. A suite of analysis experiments similar to those employed here could be performed in order to estimate the optimal averaging timescale from numerical models.

Finally, the philosophy used to create the time-mean flow estimates here has been to apply a minimum amount of mapping together with temporal averaging. A different approach would be to apply a more sophisticated mapping method such as that of e.g. Laurindo et al. (2017) involving a spatial and temporal fit, which may benefit from the inclusion of more data in this region.

# 7  Code and data availability

The various surface drifter datasets used by, and created in, this paper can be accessed as described in Table 3. The CMEMS altimeter data can be downloaded from

> http://marine.copernicus.eu/services-portfolio/access-to-products/?option=com_csw&view=details&product_id=SEALEVEL_GLO_PHY_L4_REP_OBSERVATIONS_008_047

with prior registration.

As described in the Introduction, the economic importance of the Gulf means that commercial interests often place stringent constraints on data sharing; this paper is an attempt to navigate those constraints as well as can be done, by making available to the community a derived product based in part on proprietary data that are not otherwise shared. The GulfFlow product created herein is freely available for noncommercial purposes provided one agrees not to share the data with third parties or sell products derived from it; this restricted access is a condition of the funding agent and is not negotiable. Nevertheless, its availability for noncommercial use should be sufficient for any and all academic purposes. GulfDriftersOpen, consisting of all



drifter data used herein apart from the proprietary NGOM, SGOM, and DWDE datasets, is made available without restriction. The DWDE product available is separately under the same noncommercial stipulation as for GulfFlow. For the convenience of users, a third version of GulfDrifters, GulfDriftersDWDE, is created that also contains the DWDE dataset but not the NGOM or SGOM datasets; this is available subject to the same conditions as for DWDE, at the link for DWDE given in Table 3.

All source code for data processing and figure generation associated with this paper is distributed as a part of a data analysis package for Matlab, called jLab, created and maintained by the lead author. No other special Matlab toolboxes are required. The jLab toolbox is available at the lead author's GitHub page, https://github.com/jonathanlilly, and is freely distributed under the terms of an MIT license. Figure are generated through the function `jlab_makefigs`.

## Appendix A: Numerical details

Some details regarding numerical procedures used in the analysis are briefly described in this Appendix. Here we simply point to the relevant functions in jLab, the lead author's open source data analysis toolbox for Matlab, available at https://github.com/jonathanlilly/jLab and with extensive documentation found at http://www.jmlilly.net/doc/jLab.html. Velocities are computed from hourly trajectories through a first central difference on the sphere implemented by `latlon2uv`, a routine that is also used to compute accelerations. The two-dimensional mean and histogram plots used for many of the maps are created

using `twodstats` and `twodhist`, fast and loopless functions for calculating two-dimensional statistics of large datasets. The gridded datasets were created with the function `griddrifters`. The streamline plot in Fig. 3b was created with Matlab's built-in `stream2` function. A script for making all figures associated with this paper is provided and is accessible through `jlab_makefigs`.

*Author contributions.* JML carried out the bulk of the data analysis. PPB was responsible for obtaining funding, for the planning, deployment,
and upstream processing of the SGOM and DWDE experiments, for securing and upstream processing of the NGOM dataset, for creating and archiving the final NetCDF versions of the data files, and for finding the legal pathway to make the GulfFlow dataset available. She also provided regional expertise and guidance throughout this project.

*Competing interests.* The authors declare that there are no competing interests.

*Acknowledgements.* This research, and the GulfFlow product in particular, are contributions of the Gulf of Mexico Research Consortium
(CIGoM) and were partially funded by the Mexican National Council for Science and Technology, Mexican Ministry of Energy, Hydrocarbon Fund: SENER-CONACYT/Hidrocarburos project 201441. We acknowledge Pemex's specific request to the Hydrocarbon Fund to address the environmental effects of oil spills in the Gulf of Mexico that made this project possible. The twenty-year model runs were kindly provided by CIGoM Numerical Modeling Group (CICESE, CCA-UNAM, LEGOS).


Surface Drifter Data Access for the Gulf of Mexico

| Dataset | Access | DOI |
|---|---|---|
| LATEX | https://accession.nodc.noaa.gov/download/9800141 | - |
| SCULP-I & II | Available by request. Contact PI Carter Olhmann at carter@eri.ucsb.edu. | - |
| GDP | http://www.aoml.noaa.gov/phod/dac/index.php | 10.25921/7ntx-z961 |
| HGDP | http://www.aoml.noaa.gov/phod/gdp/hourly_data.php | - |
| AOML SFP | https://www.aoml.noaa.gov/phod/sfp/data/drifter_obs.php | - |
| AOML HRD | https://www.aoml.noaa.gov/sfros/drifters/AOML_2005_hurricane_response_drifters.html | - |
| SGOM | Property of Pemex, https://www.pemex.com/en | - |
| NGOM | Property of the Woods Hole Group, https://www.horizonmarine.com/eddywatch | - |
| OCG Oildrifters | http://ocgweb.marine.usf.edu/drifter_cite.php | - |
| OCG Redtide | http://ocgweb.marine.usf.edu/drifter_cite_redtide.php | - |
| GLAD (15 min) | https://data.gulfresearchinitiative.org/data/R1.x134.073:0004 | 10.7266/N7VD6WC8 |
| Hercules | https://data.gulfresearchinitiative.org/pelagos-symfony/data/R1.x134.073%3A0012 | 10.7266/N73F4MHH |
| LASER | https://data.gulfresearchinitiative.org/data/R4.x265.237:0001 | 10.7266/N7W0940J |
| DWDE* | https://zenodo.org/record/3979964 | 10.5281/zenodo.3979964 |
| SPLASH (5 min) | https://data.gulfresearchinitiative.org/data/R4.x265.000:0074 | 10.7266/n7-0pkg-hd54 |
| AOML NSVC | https://www.aoml.noaa.gov/phod/gdp/mean_velocity.php | - |
| GulfDriftersOpen | https://zenodo.org/record/3985916 (Lilly and Pérez-Brunius, 2020a) | 10.5281/zenodo.3985916 |
| GulfFlow* | https://zenodo.org/record/3978793 (Lilly and Pérez-Brunius, 2020b) | 10.5281/zenodo.3978793 |

**Table 3.** Availability of the various drifter datasets used in this paper, together with the two datasets created herein. *Freely available for non-commercial use only.



We are grateful to Paula García, Argelia Ronquillo, Favio Medrano, and the support from Dirección de Telemática and Dirección de
Impulso a la Innovación y el Desarrollo at CICESE, as well as Lic. Omar Monroy (Mink Global) for their help in the IT and legal aspects
required for making GulfFlow available.

Horizon Marine, https://www.horizonmarine.com, was crucial during the planning, execution, and data acquisition for the SGOM drifter
program. Data from drifters deployed in US waters in support of the EddyWatch® program, https://www.horizonmarine.com/eddywatch,
have also been provided as a part of a data exchange agreement between Horizon Marine and CICESE; these comprise the NGOM dataset.

Online access to the two OCG datasets require the user to agree to include particular acknowledgement in any publication. The required
acknowledgement for the OilSpill portion of the dataset is: "Drifter data were obtained courtesy of Prof. R. H. Weisberg and the USF Ocean
Circulation Group, through a coordinated ocean observing and modeling program focusing on the West Florida Shelf (Weisberg et al., 2005).
The drifters were deployed in the eastern Gulf of Mexico region in summer 2010 as a rapid response to the Deepwater Horizon oil spill (Liu
et al., 2011, 2013a, b, c), and the data were used in monitoring the Gulf of Mexico Loop Current and the shelf circulation (Liu et al., 2013b)

and in evaluating a trajectory models (Liu and Weisberg, 2011; Liu et al., 2014)." Similarly, that for the RedTide portion is "Drifter data
were obtained courtesy of Prof. R. H. Weisberg and the USF Ocean Circulation Group for the purpose of tracking the 2012 red tide bloom
observed on the West Florida Shelf. The deployments were courtesy of the College of Marine Science, University of South Florida, Center
for Prediction of Red Tide (CPR) co-directed by Profs. R. H. Weisberg and J.J. Walsh, with assistance from the Florida Water Research
Institute and the Mote Marine Laboratory."

Finally, the first author would like to thank Phillipe Miron for help in tracking down some of the datasets.



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
