# Peer review of "A gridded surface current product for the Gulf of Mexico from consolidated drifter measurements"

_Earth System Science Data, 2020_

## Referee Comment (RC1) · Anonymous Referee #1 · 20 Oct 2020

Overview

This manuscript represents a vast amount of work to synthesize many different data sets, and is an incredible resource for future studies of circulation in the Gulf of Mexico. It should be accepted for publication.

Future users of these data, and readers in general, should pay careful consideration to the caveats noted in section 4. In particular, the drifters have various drogue depths, and some of the data sets do not include drogue presence flags. It would be very interesting to see a comparison of Fig. 2d with a similar map made from only the drogued drifters, and (for those without a drogue flag) drifters at wind speeds only below a rea-

[Figure]

sonable threshold (using winds from a satellite product). That may extend beyond the scope of this study, but would illustrate the impact of drogue loss. The authors should make clear whether the undrogued data (for those subsets with this flag) were used for Fig. 2d, Fig. 3, etc. It's clear how the data are flagged in GulfDriftersOpen, but now if the undrogued data were used here.

The authors note that simple bin average can produce spurious results due to biased sampling of particular mesoscale features during dense, short-term efforts such as LASER. They address this by averaging first in spatial bins with a one month time window, overlapping by half a month. In principle, this should be longer than the Eulerian time scale of the mesoscale variability, which the authors note later in the manuscript. It is reassuring that they applied the methodology to model output to validate it.

Specific comments:

38-40: "notwithstanding the global study of Laurindo et al. (2017) using exclusively Global Drifter Program drifters, no mean circulation maps for the Gulf of Mexico from a drifter-derived dataset have appeared in the literature since perhaps DiMarco et al.(2005) and Nowlin et al.(2001), .." Why does the map of Laurindo et al. (2017) or earlier versions cited therein, not count? This is unclear.

Section 5: it was very interesting to see the resolved vs. subgrid variance in these data. As noted by the authors, this highlights the energy levels in the submesoscale where large subgrid values are found - for example, in the Mississippi outflow. It would be interesting to examine if larger values are found in winter months, when the submesoscale may be stronger, in a future study.

Typos

30: "those derived from of dense surface drifter deployments".

---

## Referee Comment (RC2) · Anonymous Referee #2 · 11 Nov 2020

The authors carefully regrouped and processed an impressive amount of drifter data to generate two products. The first is the average state of the GoM with two different resolutions (one-quarter and one-twelfth degree) and the second product contains all drifters publicly available interpolated to hourly resolution. These products are a significant contribution to the oceanographic community studying the Gulf of Mexico (GoM), in particular for studies of the mean circulation and the interannual and seasonal variability. This manuscript is well written, the method is well explained, and the results and the method were validated using different drifter products, altimetry, and models. Thus, I recommend the publication of this manuscript after minor revisions.

[Figure]

Fig. 2 and Fig. 3a,b show a very interesting east/west connectivity in the GoM. This path extending from the LC to the Mexican coast and bounded by a westward and eastward flow in the middle of the GoM is the averaged path of the westward propagation of the Loop Current Eddies (LCE). Although this LCE path exists, I doubt it would appear in a simple average of model data, and it does not appear in the altimetry average (Fig. 2a) – even though the LCEs are resolved by altimetry. The accentuated 'LCE path' in the drifter average product is possibly due to the sampling characteristic of the drifters - which are entrained into fast currents - or due to an oversampling of the LCEs. On the other hand, the accentuated circulation allows better visualization and study of the ocean structures that would be smoothed with other tools (altimetry, model, float, etc). It would be interesting to add a couple of sentences discussing these divergences between drifter vs model and altimetry products.

l. 104. Mention that the satellite-based product only accounts for the geostrophic component of the flow.

l. 118. The lack of structures in the CMEMS compared to the NSVC product is possibly due to spatial smoothing but also because CMEMS only resolves the geostrophic velocities.

l. 160. A recirculation/closed circulation exists in the bulge of the LC, so what you are seeing in Fig. 3b is not an artifact. However, as you mentioned, the averaging method might have accentuated this recirculation.

Fig. 2d. The cyclonic feature on the northeast flank of the LC represents the strong and frequent LCFEs found in this area.

l. 556 Describe briefly the main artifacts associated with averaging drifter trajectories with altimetry data and models vs the 'truth'. Does the 'LCE path' appear in the model and altimetry 'drifter' average?

Add a sentence in section 5.4 saying that the error associated with the different size

and depth of the drogues and drogued vs undrogued drifters is not estimated.

---

## Author Comment (AC1) · 6 Jan 2021

We would like to thank the reviewer for their careful reading of our paper, and for their enthusiastic endorsement of this effort.

—Future users of these data, and readers in general, should pay careful consideration to the caveats noted in section 4. In particular, the drifters have various drogue depths, and some of the data sets do not include drogue presence flags. It would be very interesting to see a comparison of Fig. 2d with a similar map made from only the drogued drifters, and (for those without a drogue flag) drifters at wind speeds only below a reasonable threshold (using winds from a satellite product). That may extend beyond the

scope of this study, but would illustrate the impact of drogue loss. The authors should make clear whether the undrogued data (for those subsets with this flag) were used for Fig. 2d, Fig. 3, etc. It's clear how the data are flagged in GulfDriftersOpen, but now if the undrogued data were used here.

The reviewer makes a good point about the importance of the drogue flag status. In recognition of this, we have extended the source binning in GulfFlow to include three separate classes: flagged as drogued, flagged as undrogued, and no flag available. This will assist future users in discriminating the possible effect of drogue presence or loss. We have also made explicit that all data points—drogued, undrogued, or drogue status unknown—we used in making the maps. Section 2 now includes the following statement when Fig. 2c,d are introduced:

"In both cases, all available drifter data has been utilized, regardless of drogue depth and whether the drogue was estimated to be present, absent, or of unknown status."

And when Fig. 3 is introduced, we state

"The resulting mean flow estimate is shown in Fig. 3a, which again uses observations for all drogue depths and statuses. A subsequent assessment suggests this map, and the corresponding mean streamline plot in Fig. 3b, are likely not significantly influenced by bias associated with drifter sampling patterns or with potential drogue loss."

Later, in Section 5.4, we follow up this statement.

"Of the hourly data points in GulfDriftersAll, 59% have been flagged during upstream processing as being from drifters that have retained their drogue—including those from shallow drifters whose physical designs (as discussed later) make it implausible for them to lose their drogues while still transmitting—while 16% are from drifters that have lost their drogues, and 25% from drifters of an unknown drogue status. Remaking Fig. 2d using only the drogued, undrogued, or unknown status data points (not shown), one does not see evidence for substantial artifacts arising from the use undrogued drifters.

Rather, the maps from the drogued and undrogued drifters are quite comparable to each other. Similarly, remak- ing the circulation figure of Fig. 2d using only drifter drogued at shallow (1 m), intermediate (7.5 m or 15 m), or deep (45 m) depths (not shown), features that are sufficiently well sampled appear comparable regardless of the drogue depth."

"Thus, while the effects of wind slip and drogue depth variation are no doubt present, they do not appear to be major factors in shaping the mean flow maps compared with the aliasing of annual and interannual variability. An important caveat to this assess- ment is that both the drogue status and drogue depths present spatial patterns that are largely disjoint, meaning that areas that are well sampled by one class are typically less well sampled by the others. A more thorough treatment of the errors associated with drogue loss and with the use of different drogue depths would be desirable, but is outside the scope of the present work."

—38-40: "notwithstanding the global study of Laurindo et al. (2017) using exclusively Global Drifter Program drifters, no mean circulation maps for the Gulf of Mexico from a drifter-derived dataset have appeared in the literature since perhaps DiMarco et al.(2005) and Nowlin et al.(2001), .." Why does the map of Laurindo et al. (2017) or earlier versions cited therein, not count? This is unclear.

This has been rephrased as follows:

"Indeed, the only mean circulation maps for the Gulf of Mexico from drifter- derived datasets that we have identified in the literature since DiMarco et al. (2005) and Nowlin et al. (2001), at a time when the data coverage was a fraction of what it is today, are those appearing within global maps that are based exclusively on drifters from NOAA's Global Drifter Program (e.g. Lumpkin and Johnson, 2013; Laurindo et al., 2017)."

—Section 5: it was very interesting to see the resolved vs. subgrid variance in these data. As noted by the authors, this highlights the energy levels in the submesoscale where large subgrid values are found - for example, in the Mississippi outflow. It would

be interesting to examine if larger values are found in winter months, when the submesoscale may be stronger, in a future study.

We agree that this is an interesting idea, and indeed, the sub-gridscale variance maps do present seasonal variability, although we have opted not to mention this in the text as it is somewhat outside the scope.

The noted typo has been corrected.
* * *

---

## Author Comment (AC2) · 6 Jan 2021

We would like to thank the reviewer for their careful reading of our paper, for their positive comments about our work, and for drawing our attention to several important points that were deserving of more discussion.

—Fig. 2 and Fig. 3a,b show a very interesting east/west connectivity in the GoM. This path extending from the LC to the Mexican coast and bounded by a westward and eastward flow in the middle of the GoM is the averaged path of the westward propagation of the Loop Current Eddies (LCE). Although this LCE path exists, I doubt it would appear in a simple average of model data, and it does not appear in the altimetry average

[Figure]

(Fig. 2a) – even though the LCEs are resolved by altimetry. The accentuated 'LCE path' in the drifter average product is possibly due to the sampling characteristic of the drifters - which are entrained into fast currents - or due to an oversampling of the LCEs. On the other hand, the accentuated circulation allows better visualization and study of the ocean structures that would be smoothed with other tools (altimetry, model, float, etc). It would be interesting to add a couple of sentences discussing these divergences between drifter vs model and altimetry products.

We agree with the reviewer's assessment that the east/west connectivity is the primarily reflecting the averaged path of Loop Current Eddies. The possibility that the connectivity is exaggerated by oversampling of the Loop Current Eddies is a good point. We have investigated this, and added the following in the new section 5.4:

"Unlike other experiments, the drifters in the NGOM experiment were intentionally launched inside of Loop Current Eddies. This leads to the concern that the mean flow maps seen in Fig. 2d and Fig. 3 might overstate the influence of these eddies, particularly with regard to the strong east-west connectivity seen in Fig. 3b. However, remaking these figures but excluding the NGOM drifters (not shown) leads to a very similar pattern with no notable difference in the connectivity. Thus, we conclude that the connectivity seen in Fig. 3b is not an artifact of the fact that some of the drifters were intentionally launched inside of eddies."

Thus, as far as we can tell, the east/west connectivity seen in Fig. 3b is a real feature and is not an artificially enhanced by the drifter sampling.

We find that, in fact, it does appear in both the averaged CMEMS currents as well as, to a certain extent, in long-term averages from three different high-resolution numerical models that we have at our disposal. This is shown in the attached figure, which is taken from preliminary work. The weaker connectivity in the NEMO model is a consequence of the fact that the Loop Current is both too weak and located too far to the south in that model. We have therefore added the following paragraph to the end

of Section 5:

"In examining the model-derived maps, there is not evidence of major artifacts appearing in the double average over the drifter-sampled model fields in comparison to the directly-averaged model 'truth', suggesting that the drifter-inferred mean circulation seen in Fig. 2d and Fig. 3 is indeed representative of the real-world mean flow. In particular, the east-west connectivity associated with the Loop Current Eddy pathway is seen to some extent in mean streamline maps (not shown) from all three models as well as from the CMEMS data; thus we believe this feature is likely to be real. Moreover, previous studies have suggested that an average anticyclonic circulation exists in the western Gulf due both to the wind stress curl and to the westward propagation of Loop Current Eddies (e.g. Sturges and Blaha, 1976; Schmitz Jr. et al., 2005), though the relative contribution of each has yet to be determined."

While this connectivity is indeed a fascinating aspect of the circulation maps, we believe further investigation of it would be better left to a sequel.

—l. 104. Mention that the satellite-based product only accounts for the geostrophic component of the flow.

Good point, thank you. We have added the following:

"As it is based on differentiating a sea surface height anomaly measurement, this product represents only the geostrophic part of the surface currents."

l. 118. The lack of structures in the CMEMS compared to the NSVC product is possibly due to spatial smoothing but also because CMEMS only resolves the geostrophic velocities.

We believe this is probably not the main cause, but agree that it cannot be ruled out and have therefore added the following:

"The differences between the two products is likely primarily due to the larger smoothing scales in the altimeter product, although the fact that altimetry resolves only

geostrophic velocities may also play a role."

—l. 160. A recirculation/closed circulation exists in the bulge of the LC, so what you are seeing in Fig. 3b is not an artifact. However, as you mentioned, the averaging method might have accentuated this recirculation.

Thank you for pointing this out, we have removed the statement that this closed loop is an artifact. The paragraph now reads as follows.

"The streamlines corresponding to the 1/12 degree mean flow map, in Fig. 3b, emphasize the closed circulations in the Campeche Gyre and in the central and western deep Gulf. Closed time-mean circulations within the center of the Loop Current, and within a triangular region between the base of the Loop Current and Cuba, are also seen. As pointed out by an anonymous reviewer, the small closed cyclonic circulation to the north of the Loop Current most likely reflects the impact of the intense cyclonic eddies formed in the shear zone on the periphery of the Loop Current—the Loop Current Frontal Eddies (LCFEs)—that are found frequently in this area, see Le Hénaff et al. (2014) and references therein. This figure also reveals a robust east/west connectivity over the deep part of the Gulf, with streamlines reaching over some ten degrees of longitude, presumably largely reflecting the average flow associated with the westward-propagating Loop Current Eddies. Strong north-south connectivity along the western boundary is also seen."

—Fig. 2d. The cyclonic feature on the northeast flank of the LC represents the strong and frequent LCFEs found in this area.

This is a good point. We have noted this at then end of Section 2, in the paragraph quoted just above.

—l. 556 Describe briefly the main artifacts associated with averaging drifter trajectories with altimetry data and models vs the 'truth'. Does the 'LCE path' appear in the model and altimetry 'drifter' average?

This point is addressed above, in our response to the reviewer's first point regarding the LCE path.

—Add a sentence in section 5.4 saying that the error associated with the different size and depth of the drogues and drogued vs undrogued drifters is not estimated.

We have added a short new subsection, "Section 5.4, Bias considerations," before the original Section 5.4, which reads as follows.

"As discussed earlier, the GulfDrifters dataset is quite heterogeneous, with different experiments, drifter designs, and drogue depths. It is worthwhile to consider how this heterogeneity may impact the mean flow maps created using GulfFlow. Three factors that stand out are the possibility of bias due to the intentional sampling of eddies with some drifters, the possibility of artifacts arising from undrogued drifters, and the role of different drogue depths."

"Unlike other experiments, the drifters in the NGOM experiment were intentionally launched inside of Loop Current Eddies. This leads to the concern that the mean flow maps seen in Fig. 2d and Fig. 3 might overstate the influence of these eddies, particularly with regard to the strong east-west connectivity seen in Fig. 3b. However, remaking these figures but excluding the NGOM drifters (not shown) leads to a very similar pattern with no notable difference in the connectivity. Thus, we conclude that the connectivity seen in Fig. 3b is not an artifact of the fact that some of the drifters were intentionally launched inside of eddies."

"Of the hourly data points in GulfDriftersAll, 59% have been flagged during upstream processing as being from drifters that have retained their drogue—including those from shallow drifters whose physical designs (as discussed later) make it implausible for them to lose their drogues while still transmitting—while 16% are from drifters that have lost their drogues, and 25% from drifters of an unknown drogue status. Remaking Fig. 2d using only the drogued, undrogued, or unknown status data points (not shown), one does not see evidence for substantial artifacts arising from the use undrogued

drifters. Rather, the maps from the drogued and undrogued drifters are quite compara-
ble to each other. Similarly, remaking the circulation figure of Fig. 2d using only drifter
drogued at shallow (1 m), intermediate (7.5 m or 15 m), or deep (45 m) depths (not
shown), features that are sufficiently well sampled appear comparable regardless of
the drogue depth."

"Thus, while the effects of wind slip and drogue depth variation are no doubt present,
they do not appear to be major factors in shaping the mean flow maps compared with
the aliasing of annual and interannual variability. An important caveat to this assess-
ment is that both the drogue status and drogue depths present spatial patterns that
are largely disjoint, meaning that areas that are well sampled by one class are typically
less well sampled by the others. A more thorough treatment of the errors associated
with drogue loss and with the use of different drogue depths would be desirable, but is
outside the scope of the present work."

We have also changed the first sentence of Section 5.5 (previously 5.4) to clarify the
type of errors that will be estimated:

"While the true mean flow in the Gulf of Mexico is unknown, it is nonetheless possible to
quantify errors in estimating the mean flow arising from the interaction of the averaging
methods with the drifter space/time sampling pattern."
* * *
[Figure]

Fig. 1. Mean streamlines from models and from altimetry

---

## Author Response (AR1)

The reponses to the reviewers have been detailed in our public responses to their comments. In particular, we have:

1. Added an additional drogue status option, in response to an issue brought up by Reviewer 1, see lines 500--505 of gulfdrifters-diff, and discussed drogue status futher on lines 422--428.
2. Addressed a few small clarifications addressed by Reviewer 1
3. Added a new Section 5.4 in response to a question by Reviewer 2 regarding potential bias sources
4. Discussed in more detail the streamline pattern to address an issue raised by Reviewer 2, see lines 170--180 and 629--636 of gulfdrifters-diff
5. Addressed a number of minor points raised by Reviewer 2
6. Included an additional year of Global Drifter Program data, changing the overall span of the product from 27 to 28 years
7. Made a minor change to an editing criterion, leading to small changes in Table 1.
8. Remade Table 2 to account for the changes to the dataset
9. Corrected some typos, and added minor clarifcations throughout

[revised manuscript text omitted]